# Boosting Verifiable Industrial Code Generation by Reliable Task Generation at Scale

## Abstract

Recent advances in LLMs and industrial copilots (e.g., from Siemens, Rockwell and Schneider) have the potential to transform the way control engineers program. Generating verifiable industrial code (e.g., free from both syntactic and logical errors) via LLMs remains inherently challenging, however, due to strict safety constraints and the intolerance for logical failures. The closed-source nature and scarcity of data for Industrial Control System (ICS) programming tasks exacerbate this difficulty, preventing LLMs from reaching their transformative potential in ICSs. To address this critical gap, we introduce PLC-Spec-Syn, the first evolutionary framework to generate high-fidelity PLC programming tasks. Each task consists of a detailed **specification**—a structured, natural language engineering document—and its corresponding verified PLC code. The core idea is to guide LLM-based task generation (specification–code pair) with practical industrial engineering principles through a multi-axis evolutionary process considering six dimensions: functionality, safety, performance, maintenance, interoperability, and contextual complication. To ensure data quality, each generated specification–code pair will undergo rigorous auditing including compilation check and formal verification of semantic consistency between the specification and the code. The whole process yields PLC-Spec-Code, the first large-scale corpus of 11,669 PLC programming tasks[1] with strict quality control. Besides, PLC-Spec-Code has 84.3% syntactic diversity, substantially exceeding that of existing corpus like OSCAT (29.2%). Importantly, fine-tuning multiple (code) LLMs using our corpus improves their performance on verifiable PLC code generation in unseen tasks by an average of 16.4% compared to the previous models, confirming the effectiveness of our task generation approach and the practical usefulness of our corpus.

## 1 Introduction

Industrial control systems (ICS) execute safety-critical programs on programmable logic controllers (PLCs), typically written in IEC 61131-3 (International Electrotechnical Commission, 2013a) Structured Text (ST) code. Recently, there is a surge in LLM-based industrial copilots (first introduced by Siemens in July 2024), which have the potential to transform the way how control engineers program. While existing code LLMs or some general-purpose LLMs can achieve strong performance on mainstream programming languages, such as C and Python (Zhang et al., 2024), their performance often degrade on industrial programming languages like ST with strict real-time and safety constraints (Roziere et al., 2023; Li et al., 2022; Zhang et al., 2024; Xu et al., 2024; Joel et al., 2024; Bassamzadeh & Methani, 2024). For instance, LLMs may generate programs that compile correctly yet violate implicit permissions, misuse timers or counters, or fail under realistic interlocks (Liu et al., 2024; Fakih et al., 2024).

In terms of LLM-based industrial code generation, one fundamental challenge is the lack of high-quality training data. Although open resources, such as OSCAT (oscat.de, 2024-11-03), are valuable, they are narrow in scope, small in scale and primarily offer function code blocks instead of diverse, end-to-end programming tasks. This limitation has been widely acknowledged as a bottleneck for pushing the boundary of verifiable code generation in industrial control applications (Haag et al., 2025). While existing generic data synthesis paradigms, such as Self-Instruct (Wang et al., 2022) and

---

[1]In comparison, the largest prior corpus OSCAT has 718 tasks.

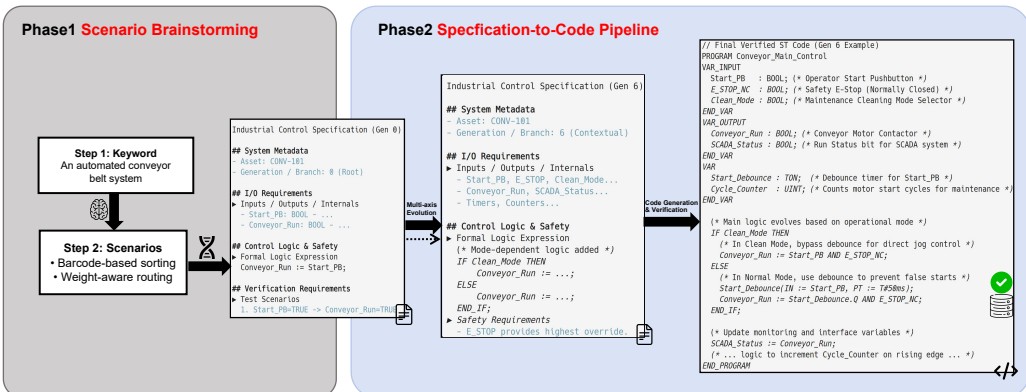

Figure 1: The end-to-end workflow of our PLC-Spec-Syn framework. The process starts from a high-level **Keyword**, which is brainstormed into diverse **Scenarios**. Each scenario is then formulated into a detailed, structured **Specification** (Gen 0). This specification undergoes principled, multi-axis **Evolution** to a more complex state (e.g., Gen 6). Finally, the evolved specification is used to generate verified ST **Code**, and the resulting high-fidelity pair is added to the PLC-Spec-Code corpus. The full specification template is detailed in Appendix U.

Evol-Instruct (Li et al., 2024), fail to meet stringent industrial requirements without incorporating industrial engineering principles.

**PLC-Spec-Syn**. To address this critical need for a high-quality industrial programming corpus, we present PLC-Spec-Syn, the first LLM-based evolutionary PLC programming task generation framework. The entire pipeline, from initial task conception to final code generation, is summarized in Figure 1. Our framework embeds industrial control principles to guide LLMs through this complete, end-to-end process for generating high-fidelity tasks. Specifically, PLC-Spec-Syn operationalizes established practices in industrial control (e.g., phased development, safety precedence) into a set of rules for staged, multi-axis evolution for automatic task generation at scale. This evolutionary process, as detailed in Figure 2, begins with 111 carefully crafted curriculum-grounded seed keywords covering a broad range of industrial programming scenarios, and then performs staged, safety-aware evolution across six orthogonal branches: functionality, safety, performance, maintenance, interoperability, and contextual complication. Through this rigorous process, we obtain PLC-Spec-Code, the first formally verified instruction–code corpus for PLC programming, consisting of 11,669 verified pairs.

In summary, we make the following main contributions:

- We introduce PLC-Spec-Syn, the first automatic industrial task generation framework for encoding industrial engineering principles into a multi-axis instruct-code corpora generation process, enabling controllable, transparent, and safety-aware LLM-based industrial programming task synthesis at scale.
- We release PLC-Spec-Code, a large-scale (11669 V.S. prior largest 718) verified instruction–code corpus for industrial programming, which is shown useful to significantly enhance the verifiable industrial code generation ability of existing models by simple fine-tuning. The corpus also enables more comprehensive evaluation and benchmarking of the LLM-based PLC code generation field.
- We release the PLC-Spec-Code dataset as open-source, available at `https://anonymous.4open.science/r/ICLR_-0CF4`.

## 2 RELATED WORK

### 2.1 INDUSTRIAL CONTROL SYSTEMS AND ST CODE

Industrial Control Systems (ICS) form the backbone of automation in critical infrastructure, where ruggedized computers known as Programmable Logic Controllers (PLCs) execute real-time control logic (Petruzella, 2016). These devices are programmed using languages defined by the IEC 61131-3 standard, among which Structured Text (ST) is a high-level, text-based language essential for implementing complex algorithms. Unlike in general-purpose computing, ST code directly translates

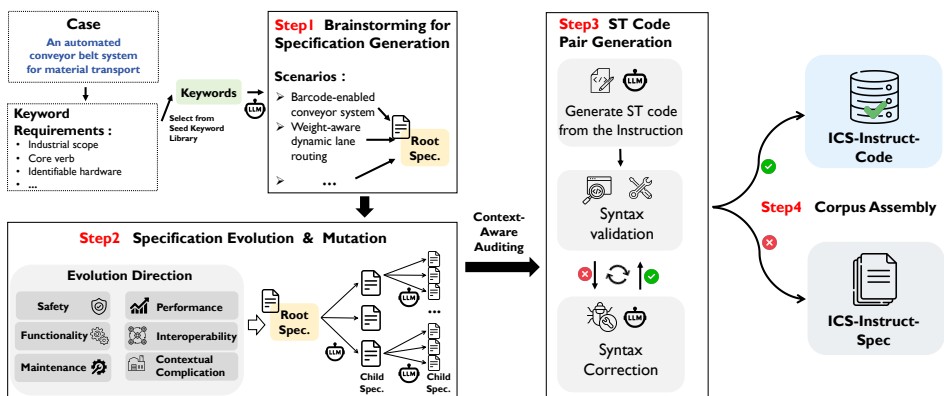

Figure 2: Overview of PLC-Spec-Syn. From each *root specification* (root spec.), the Step2 process branches through six generations. Each generation applies one enhancement branch (functionality, safety, performance, maintenance, interoperability, or contextual complexity) under stage-bounded rules.

digital commands into physical actions in high-stakes environments. Consequently, a single semantic error—a logical flaw that does not prevent compilation—can lead to catastrophic equipment failure, production loss, and severe threats to human safety (International Electrotechnical Commission, 2010). This extreme sensitivity to logical correctness highlights the urgent need for diverse and semantically-grounded corpora to train generative models capable of producing verifiably safe and reliable control programs.

## 2.2 LLM-BASED CODE GENERATION

Large Language Models (LLMs) (Austin et al., 2021) have gained widespread adoption in various domains of coding tasks, with notable potential showcased in code completion (Chen et al., 2021), automated program repair (Fan et al., 2023), program testing (Vikram et al., 2023), program proof (Yang et al., 2023), and repository-level programming (Liu et al., 2023).

**LLM-based PLC Code Generation.** To enhance PLC code generation, LLM4PLC (Fakih et al., 2024) adopts an iterative pipeline guided by LLMs and supplemented with external verification tools. Meanwhile, Agents4PLC (Liu et al., 2024) and AutoPLC (Yang et al., 2024) introduce multi-agent frameworks that automate both the generation and verification of PLC code. Such diverse approaches underscore the rising interest in applying LLMs to PLC code development in industrial contexts.

## 2.3 DATA AUGMENTATION

Conventional augmentation—syntax-preserving rewrites (FOR → WHILE), back-translation, adversarial perturbations—can increase data volume but rarely creates *new* industrial tasks or broadens the semantic envelope of control problems (Hort et al., 2025; Wang et al., 2023; Chen & Lampouras, 2023; Yasunaga & Liang, 2021; Richter & Wehrheim, 2022; Boudribila et al., 2025). In safety-critical ICS, such rewrites risk superficially diverse yet semantically brittle samples (Honovich et al., 2022; Ding et al., 2024). Generic instruction-synthesis paradigms (Self-Instruct, Evol-Instruct) are powerful but lack mechanisms to *systematically* grow tasks along orthogonal engineering axes (e.g., safety vs. performance) (Wang et al., 2022; Li et al., 2024). However, PLC-Spec-Syn evolves task specifications under structured rules to generate realistically complex scenarios across multiple engineering dimensions.

## 2.4 EVOLUTIONARY PRINCIPLES AND QUALITY–DIVERSITY

PLC-Spec-Syn is inspired by evolutionary computation and quality–diversity (QD), which jointly optimize for performance and novelty across structured search spaces (Gravina et al., 2019). Recent studies draw parallels between iterative LLM generation and population-based search (Chao et al., 2024), and hybrid methods that blend genetic programming with LLM priors show promise on complex programming tasks (Guadalupe Hernandez et al., 2025). We operationalize this synergy at the *specification* level: staged mutations with bounded increments expand IEC 61131-3 coverage

(e.g., CASE/ENUM, TON/CTU, ARRAY/STRUCT) while preserving safety precedence and local-control authority.

## 3 METHODOLOGY

PLC-Spec-Syn proposes a high-fidelity instruction–code generation pipeline, consisting of two primary phases: 1) specification seeding and evolution, and 2) a multi-stage validation and curation workflow. Then we obtain PLC-Spec-Code, a formally verified instruction–code corpus for ICS.

### 3.1 THE PLC-SPEC-SYN FRAMEWORK

PLC-Spec-Syn is a generative framework that iteratively evolves *industrial control* task specifications using large language models (LLMs), as shown in Figure 2. Starting from a minimal seed specification, the system produces increasingly complex, safety-aware specifications through small, controlled modifications. Each modification targets an orthogonal aspect (branch) of control logic—adding functionality, safety constraints, performance optimizations, maintenance features, interoperability hooks, or contextual complications—while preserving all prior behavior. Generations are grouped into early, mid, and late stages to ensure gradual complexity growth. The entire end-to-end data generation pipeline is formally detailed as pseudocode in 1 in the appendix.

**Terminology and counting conventions.** We distinguish four levels in our data generation pipeline: (i) we begin with 111 curriculum-grounded *keywords* $K$ (see Appendix F for selection criteria and examples), (ii) for each keyword $k \in K$, we generate $S = 4$ diverse *scenarios* (see Appendix E for the brainstorming methodology), (iii) after de-duplication, these scenarios are consolidated into $R$ *root specifications* ($|R| = 347$) that serve as the Evolutionary Generation 0 (Gen 0) starting points, and (iv) each root expands into a six-generation tree of *child specifications* via PLC-Spec-Syn. Formally:

$$K \xrightarrow{\text{brainstorm } (S=4)} \text{scenarios} \xrightarrow{\text{dedup}} R \text{ (root specs, } |R|=347) \xrightarrow{\text{6-gen evolution}} \text{child specs.}$$

We consistently use **keyword / scenario / root spec / child spec** throughout.

Table 1: Entities and counts used in PLC-Spec-Syn.

| Term | Definition | Count |
|------|------------|-------|
| Keyword ($K$) | Curriculum-grounded seed prompts | $|K| = 111$ |
| Scenario | Brainstormed variants per keyword | $S = 4$ per keyword (444 total) |
| Root spec ($R$) | Deduplicated scenario set (tree roots) | $|R| = 347$ |
| Child spec | Nodes produced by 6-gen evolution | per-root tree (kept in full) |

### 3.2 PHASE 1: SPECIFICATION SEEDING AND EVOLUTION

**Task Seeding and Diversification.** We began with a curated corpus of 111 *seed keywords* designed to reflect canonical industrial automation challenges, spanning major sectors as detailed in Table 2. To expand these keywords into a diverse task set, we used a two-stage generate-and-filter process. First, an LLM brainstorming prompt produced $S=4$ distinct *scenarios* for each keyword, yielding 444 candidates. Second, we applied programmatic de-duplication using a hybrid similarity score (semantic, lexical, and metadata). Pairs above a threshold were clustered, and one representative was retained. This process reduced the pool to $|R| = 347$ high-diversity *root specifications*, which served as the starting points for evolution. Appendix F illustrates the breadth of the 111 seed keywords with a representative sample categorized by industrial domain.

Table 2: Distribution of the 111 seed *keywords* across major industrial domains.

| Industrial Domain | Key Areas Covered | No. of Keywords |
|-------------------|-------------------|-----------------|
| **Manufacturing & Assembly** | Robotic Welding, Pick & Place, Semiconductor Fab, Automotive | 28 |
| **Material Handling & Logistics** | Conveyors, Palletizing, AGVs, Sorting, Packaging, Warehousing | 18 |
| **Process Control** | Chemical Batching, Food & Beverage, Pharma, Water Treatment | 24 |
| **Energy, Resources & Utilities** | Power Generation, Renewables, Mining, Oil & Gas, Water Management | 12 |
| **Infrastructure & Agriculture** | HVAC, Elevators, Smart Buildings, Irrigation, Climate Control | 16 |
| **Safety & Monitoring** | Gas & Fire Alarms, Leak Detection, Predictive Maintenance, SCADA | 13 |

**Multi-Branch Evolution Strategy.**  With the 347 root specifications established, we apply the principled, guided multi-axis evolution. The process uses six orthogonal branches—**functionality**, **safety**, **performance**, **maintenance**, **interoperability**, and **contextual complication**—derived from industrial standards like IEC 61508. To ensure pedagogically sound complexity growth, we implemented Smooth Evolution Pacing (SEP), which constrains modifications to be cumulative over six generations(Appendix H). The evolution is structured into early, mid, and late stages, with branches sampled probabilistically at each step to create diverse and realistic development paths. The detailed stage-dependent probabilities for branch selection are provided in Appendix G.

**Evolutionary Pipeline Execution.**  Following the multi-branch evolution strategy, we evolved our 347 `root specifications` over six generations. This process yielded a raw candidate pool of **20,362** specifications.

## 3.3 PHASE 2: MULTI-STAGE VALIDATION AND CORPUS CURATION

To ensure the quality and validity of the generated data, we implemented a rigorous, multi-stage filtering pipeline, summarized in Table 12.

**Context-Aware Specification Auditing.**  We subjected all **20,362** raw specifications to a context-aware, multi-agent audit. A committee of three specialist LLM agents evaluated each specification, with prompts dynamically contextualized based on the specification's evolutionary stage. A specification was approved only if its context-aware average score was $\geq 4.0$. This audit filtered the pool down to **18,247** high-quality and evolutionarily valid specifications.

**Human Evaluation.**  To provide a gold-standard assessment, we conducted a user study with an evaluation panel consisting of five graduate students (2 MSc, 3 PhD) specializing in industrial automation and one industrial engineer with five years of professional experience. The study involved two tasks: rating the quality of 1,000 randomly sampled specifications and evaluating the logical coherence of 30 complete evolutionary lineages. The detailed study methodology is provided in Appendix P.

**Code Generation and Syntactic Validation.**  We prompted our generative engine (DeepSeek-V3.1, configured with temperature $T = 0.1$ for deterministic output) to produce Structured Text (ST) code for the **18,247** approved specifications. The code was compiled with `matiec`, with up to 3 generation attempts on failure. This step served as a realizability check and syntactic validation, yielding **14,292 pairs** where the code was syntactically correct and successfully compiled. A detailed, generation-by-generation breakdown of these statistics is presented in Appendix L.

**Formal Verification of Semantic Equivalence.**  To establish semantic alignment with mathematical certainty, we adopted a formal verification approach using the `nuXmv` model checker (Fakih et al., 2024). Crucially, to guarantee the correctness of the verification properties, we employ a **neuro-symbolic approach**: our task generation phase forces the LLM to output a structured "Formal Logic Expression" block (see Appendix U), which is then translated into LTL/CTL formulas via a **deterministic parser**, decoupling logic extraction from generation. Human evaluation by domain experts on a stratified sample confirms that this approach achieves **91.5%** semantic alignment with the design intent, compared to 62.0% for direct LLM generation. This process translates both the specification and the code into temporal logic formulas, formally proving their equivalence. This final, stringent quality gate was applied to all 14,292 compiling pairs, filtering the corpus down to **11,669 formally verified pairs**. To further validate the quality of this final set, we conducted a human evaluation where three graduate students rated a random sample of 500 pairs on their instruction-code alignment. The results confirmed that 100% of the inspected pairs from the verified set achieved an "excellent match" rating. The detailed statistics for the entire filtering funnel are presented in Appendix L.

## 3.4 THE PLC-SPEC-CODE CORPUS

The two-phase generation and validation process resulted in the PLC-Spec-Code. The primary corpus, used for our downstream experiments, contains the **11,669 instruction-code pairs that successfully passed compilation and formal verification**. The generation of a single successful instruction-code pair required between 1,000 and 8,000 tokens, averaging 6,000, with token consumption increasing in later generations due to the cumulative growth in task complexity. A secondary corpus contains high-quality specifications for which automated code generation either

failed compilation or could not be formally verified, providing challenging cases for future program synthesis research.

# 4 EXPERIMENTS AND EVALUATION

This section details the experimental setup and results. We aim to answer the research questions (RQs) through quantitative metrics.

**RQ1** What is the semantic diversity and coverage of the code corpus generated by PLC-Spec-Syn?

**RQ2** How does fine-tuning on PLC-Spec-Code affect model performance on industrial code generation tasks?

**RQ3** How does PLC-Spec-Syn's structured process compare in quality and effectiveness to alternative synthesis paradigms?

## 4.1 EXPERIMENTS SETUP

**Datasets.** We use several standard industrial benchmarks for training, comparison, and evaluation.

- *OSCAT Library*(oscat.de, 2024-11-03): A key baseline corpus from a community-driven, open-source project, containing 718 function blocks. While extensive, its samples are primarily isolated components with limited diversity in complex control flow.

- *Four Downstream Evaluation Datasets:* We use four industrial benchmarks, three of which are established testbeds used to validate the state-of-the-art AutoPLC framework (Yang et al., 2024): 1) OSCAT Library, 2) the official Siemens LGF Library (Siemens, 2024), 3) the complex Siemens Competition Dataset (biendata.xyz, 2024), and 4) the LLM-oriented Agents4PLC Corpus Liu et al. (2024). A detailed description of each evaluation set is provided in Appendix Q.

- **PLC-Spec-Code (Ours)**: The primary training set, containing 11,669 formally verified instruction-code pairs generated by our PLC-Spec-Syn framework.

**Models and Fine-Tuning Setting.** We primarily used four kinds of models, including:

- *Generative Model*: We use DeepSeek-V3.1, configured with a high temperature ($T = 0.6$) for diverse specification evolution and a low temperature ($T = 0.1$) for deterministic code translation (DeepSeek-AI, 2024).

- *Auditor Consortium*: An evaluation committee of three distinct LLMs (GLM-4.5, Kimi-K2, and Qwen3-235B) assesses all specifications for quality. The detailed audit methodology is provided in Appendix O.

- *Validation Tools*: The technical validation suite includes the `matiec` compiler for syntactic checking and the `nuXmv` model checker for formal semantic verification (see Appendix J).

- *Fine-tuned Models*: For downstream evaluation, we fine-tune several open-source models, including `Qwen2.5-Coder-1.5B-Instruct`, `Qwen2.5-Coder-7B-Instruct`, and `Llama-3.2-1B-Instruct`.

**Evaluation Protocol.** To ensure a fair and consistent comparison, all models—both the base open-source models and their fine-tuned counterparts, as well as all proprietary baselines—were evaluated under an identical, strict **zero-shot** protocol. Each model was given the exact same minimal prompt containing only the task specification, without any examples (few-shot), chain-of-thought instructions, or other engineered guidance. This methodology isolates the impact of the knowledge gained during fine-tuning on PLC-Spec-Code, rather than evaluating prompt engineering techniques. Our primary goal is therefore to demonstrate the utility of our dataset for improving a model's foundational capabilities, not to determine the absolute best-performing model under all possible conditions.

**Baselines.** To validate the superiority of our PLC-Spec-Syn framework, we compare it against five alternative data generation strategies. For a fair comparison, each baseline method was used to create a 500-sample corpus. A detailed description of the methodology and prompt design for each baseline is provided in Appendix S.

- *Traditional Augmentation (TA-Corpus)*: Represents a **code-level evolution** approach. It applies simple, syntax-preserving rewrites (e.g., variable renaming, loop swaps) to the programs in a 500-sample stratified subset of our own generated corpus to test the value of superficial syntactic variety.

- *Self-Instruct Proxy (SI-Proxy)*: Mimics the Self-Instruct paradigm by generating new, distinct industrial tasks from our root specifications, representing a standard instruction synthesis approach.

- *Evol-Instruct Proxy (EI-Proxy)*: Simulates Evol-Instruct by using a generic "make it more complex" prompt on our root specifications to test the effectiveness of unguided complexity growth.

- *One-Shot Expert Prompt (OSEP-Corpus)*: A strong non-evolutionary baseline that uses a single, highly-detailed "expert prompt" to generate a complex, multi-feature specification in one step, integrating aspects like safety, performance, and maintenance simultaneously.

- *Modular Composition (MC-Corpus)*: Simulates a bottom-up engineering approach by using an LLM to combine simpler "Atomic" (Gen 0-1) and "Developed" (Gen 2-3) specifications into more complex, integrated tasks.

**Metrics.** We use the following key metrics:

1) *Compilation Success Rate (CSR):* This is our primary metric for downstream performance evaluation (RQ2 & RQ3). It measures an LLM's ability to generate syntactically valid code for a given task specification from an unseen industrial test set.

2) *Syntactic Coverage:* To measure corpus diversity and coverage (RQ1), we calculate the percentage of syntactic tokens used in the generated code against a pre-defined 89-token ST syntax taxonomy. This allows for a direct comparison of linguistic variety against baselines.

3) *Complexity Scores:* To quantify the complexity of our generated tasks (RQ1 & RQ3), we use two scores. The **Specification Richness Score** evaluates the complexity of the natural language instruction based on its technical keywords and I/O variables. The **Code Complexity Score** assesses the structural and logical intricacy of the generated code, incorporating metrics like *Cyclomatic Complexity*.

### 4.2 RQ1: CORPUS DIVERSITY, COMPLEXITY, AND COVERAGE

To answer RQ1, we first analyzed the syntactic diversity of the 11,669 pairs in PLC-Spec-Code against the OSCAT library, measuring coverage of an 89-token ST syntax taxonomy (see Appendix M). As shown in Table 3, our corpus demonstrates a clear curriculum effect, with overall syntax coverage growing to **84.3%**, nearly triple OSCAT's 29.2%. The most significant gain is in `Control Flow Statements`, where our coverage reaches **95.2%** compared to OSCAT's 9.5%, filling a critical gap left by existing resources.

Table 3: Progression of syntax coverage (%) against our 89-token ST taxonomy. Our corpus shows significantly higher coverage in key areas like Control Flow.

| Feature Family | Gen 0 (Base) | Gen 2 (Early Stage) | Gen 4 (Mid Stage) | Gen 6 (Late Stage) | OSCAT |
|---|---|---|---|---|---|
| DataTypes (Elem. & Comp.) | 6.4% | 32.1% | 42.9% | 71.4% | **85.7%** |
| Control Flow Statements | 0.0% | 28.6% | 42.9% | **95.2%** | 9.5% |
| Standard Function Blocks | 0.0% | 55.6% | **77.8%** | **77.8%** | 22.2% |
| Operators (All Types) | 6.7% | 50.0% | 53.3% | **90.0%** | 14.3% |
| Overall Syntax Coverage | 9.0% | 41.6% | 53.9% | **84.3%** | 29.2% |

Beyond syntactic coverage, we analyzed the growth in richness and complexity across generations using a suite of quantitative metrics (detailed in Appendix K). Figure 3 visually summarizes this curriculum effect. The analysis reveals a strong, monotonic increase in both the median specification richness and the median code complexity with each generation. Furthermore, the expanding min/max range (the whiskers) demonstrates that later generations not only contain more complex tasks but also a wider diversity of complexity. This confirms that PLC-Spec-Syn successfully generates a curriculum of progressively more challenging and varied tasks.

**Independent Diversity Analysis.** To address potential bias in standard-based metrics, we evaluated diversity using four independent metrics:

- **Lexical Diversity:** Our dataset contains **3.2×** more unique 4-grams than the OSCAT baseline, indicating a wider vocabulary of code patterns and local phrasing.
- **Structural Diversity:** The average Tree Edit Distance (TED) between pairs of programs in our dataset is **0.70**, compared to 0.34 in OSCAT, proving our programs are structurally far less repetitive.
- **Semantic Diversity:** We measured the Type-Token Ratio (TTR) to assess identifier richness. Our dataset scored **1.5×** higher than the baseline, reflecting context-specific naming (e.g., `Conveyor_Jam_Sensor`) rather than generic placeholders.
- **Task Rarity:** In a blind expert evaluation, **32%** of our late-stage tasks were classified as "Rare/Complex" scenarios (specific to complex production) that are typically absent from standard libraries like OSCAT.

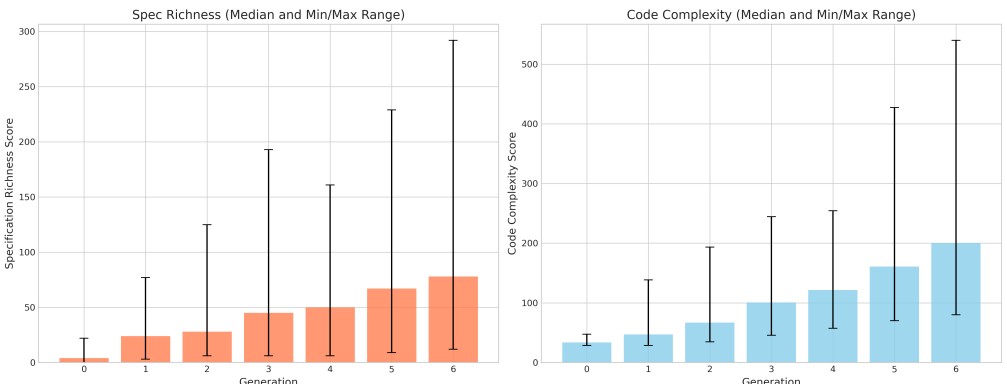

Figure 3: Distribution of Richness and Complexity Scores Across Generations. The height of each bar represents the median score, while the whisker shows the full min/max range. Both specification richness and code complexity show a clear, steady increase, validating the curriculum effect of our evolutionary process.

### 4.3 RQ2: EFFECTIVENESS OF DOMAIN KNOWLEDGE VIA FINE-TUNING AND RAG

To answer RQ2, we conducted a comprehensive evaluation to assess how domain-specific knowledge—injected via **Supervised Fine-Tuning (SFT)**, **Prompt Engineering (PE)**, and **Retrieval-Augmented Generation (RAG)**—affects model performance across different model scales and architectures.

**Supervised Fine-Tuning vs. Prompt Engineering.** We evaluated three open-source models (`Qwen2.5-Coder-1.5B/7B-Instruct` and `Llama-3.2-1B-Instruct`) under three settings: (1) **Base (Zero-shot)**, (2) **Base + PE**, and (3) **Fine-tuned (FT)** on PLC-Spec-Code.

**Prompt Design for PE:** To ensure a strong baseline, we designed a sophisticated "Expert Prompt" incorporating three strategies:

- **Role-Playing:** "You are an expert PLC programmer specializing in IEC 61131-3 Structured Text (ST)..."
- **One-Shot Example:** We provided a canonical `Simple_Motor_Starter` example (Input: `Start_PB`, Output: `Motor_Run`) to demonstrate correct syntax and variable declaration structure.
- **Chain-of-Thought (CoT):** We explicitly instructed the model to "think step by step... First, analyze variables... Second, break down logic... Finally, generate code."

The comparative results are presented in Table 4. Key observations include:

- **FT dominates PE:** While Prompt Engineering consistently improves over the zero-shot baseline (e.g., Qwen-1.5B improves from 12.7% to 18.5% on OSCAT), it strictly underperforms Fine-

Tuning (which reaches 30.1%). This indicates that the complexity of industrial logic requires weight updates rather than just context guidance.

- **Scale Efficiency:** Notably, the fine-tuned small model (`Qwen-1.5B-FT`) often outperforms the prompt-engineered larger model (`Qwen-7B + PE`) on challenging tasks (e.g., Agents4PLC: 61.0% vs 57.8%), highlighting the efficiency of domain-specific training.

Table 4: Comprehensive comparison of Base, Prompt Engineering (PE), and Fine-Tuning (FT) across all models and datasets. FT consistently yields the highest performance gains.

| Test Set | Qwen2.5-Coder-1.5B-Instruct | | | Qwen2.5-Coder-7B-Instruct | | | Llama-3.2-1B-Instruct | | |
|---|---|---|---|---|---|---|---|---|---|
| | Base | Base + PE | Fine-tuned | Base | Base + PE | Fine-tuned | Base | Base + PE | Fine-tuned |
| **OSCAT (718)** | 12.7% | 18.5% | **30.1%** | 15.0% | 20.3% | **31.6%** | 2.2% | 3.3% | **17.0%** |
| **LGF (151)** | 4.6% | 6.6% | **24.5%** | 13.2% | 17.9% | **25.2%** | 2.0% | 3.3% | **18.5%** |
| **Competition (45)** | 13.3% | 20.0% | **26.6%** | 17.8% | 24.4% | **33.3%** | 0.0% | 0.0% | **11.1%** |
| **Agents4PLC (187)** | 43.3% | 49.7% | **61.0%** | 51.3% | 57.8% | **76.5%** | 2.7% | 4.3% | **26.7%** |

**Effectiveness of RAG** We conducted a Retrieval-Augmented Generation (RAG) experiment using **DeepSeek-V3.1** model. We compared its zero-shot performance against retrieving from the existing **OSCAT** library versus retrieving from our **PLC-Spec-Code**. **Results:** As shown in Table 5, using PLC-Spec-Code as the knowledge base dramatically increased the overall success rate on the combined test set (383 tasks) from 51.7% to **70.0%**. This represents a **35.4% relative improvement** over the OSCAT-RAG baseline (56.7%). The massive gain confirms that providing high-quality, domain-specific data effectively lowers the barrier to verifiable industrial code generation.

Table 5: RAG Performance Analysis using DeepSeek-V3.1. Retrieving from PLC-Spec-Code significantly outperforms retrieving from the existing OSCAT library, proving the value of our dataset's diversity and quality.

| Dataset | Zero-shot | RAG (w/ OSCAT) | RAG (w/ PLC-Spec-Code) |
|---|---|---|---|
| LGF (151) | 29.8% | 39.7% | **55.0%** |
| Competition (45) | 24.4% | 22.2% | **44.4%** |
| Agents4PLC (187) | 75.9% | 78.6% | **88.2%** |
| **Overall (383 tasks)** | 51.7% | 56.7% | **70.0%** |

Table 6: Comparison of functional verification performance. Fine-tuning on PLC-Spec-Code not only improves compilation (CSR) but drastically increases the Verified Success Rate (VSR) and Semantic Consistency (VSR/CSR), indicating the model learns valid control logic rather than just syntax.

| Model | Setting | Compilation Success Rate (CSR) | Verified Success Rate (VSR) | Semantic Consistency (VSR/CSR) |
|---|---|---|---|---|
| Qwen-1.5B | Base | 18.5% | 6.9% | 37.3% |
| | Fine-tuned | **33.9%** (+15.4%) | **26.5%** (+19.6%) | **78.2%** (+40.9%) |
| Qwen-7B | Base | 24.3% | 9.1% | 37.4% |
| | Fine-tuned | **41.6%** (+17.3%) | **33.7%** (+24.6%) | **81.0%** (+43.6%) |
| Llama-1B | Base | 1.7% | 0.6% | 35.3% |
| | Fine-tuned | **18.3%** (+16.6%) | **14.3%** (+13.7%) | **78.1%** (+42.8%) |

**Verification Success Rate (VSR) and Functional Correctness.** Beyond compilation, we assessed functional correctness using formal verification via `nuXmv`. As shown in **Table 6**, fine-tuning significantly improves the **Verified Success Rate (VSR)**. For instance, the VSR for Qwen-1.5B increased from 6.9% (Base) to **26.5% (Fine-tuned)**. Crucially, the **Semantic Consistency** (the ratio of VSR to CSR) improved dramatically from ~37% to ~80% across all models. This confirms that PLC-Spec-Code enables models to learn genuine functional logic, ensuring that syntactically valid code is also semantically correct.

**Comparison with LLM4PLC and RAG.** We compared our method against the **LLM4PLC** framework using the powerful DeepSeek-V3.1. While LLM4PLC achieved 28.8% compilation

Table 7: Detailed complexity analysis across all generation methods. PLC-Spec-Syn (Overall Avg.) produces a corpus with significantly higher specification richness and code complexity than the baselines, while maintaining high semantic similarity.

| Data Generation Method | Avg. Spec Richness Score | Avg. Code Complexity Score | Avg. Cyclomatic Complexity | Avg. Semantic Similarity | Code Gen. Success Rate (%) |
|---|---|---|---|---|---|
| *Baseline Methods* | | | | | |
| Traditional Augmentation (Source) | 58.46 | 138.97 | 10.85 | 0.8393 | 100% |
| Traditional Augmentation (Augmented) | 58.46 | 138.97 | 10.82 | 0.8375 | 100% |
| Self-Instruct Proxy (SI) | 22.12 | 64.59 | 3.61 | 0.8552 | 98.5% |
| Evol-Instruct Proxy (EI) | 25.72 | 152.68 | 10.22 | 0.8601 | 89.0% |
| One-Shot Expert Prompt (OSEP) | 69.99 | 242.84 | 18.89 | 0.8588 | 40.3% |
| Modular Composition (MC) | 48.43 | 345.24 | 23.14 | 0.8578 | 55.0% |
| *PLC-Spec-Syn (Our Method) by Generation* | | | | | |
| PLC-Spec-Syn (Gen 0) | 4.57 | 32.77 | 1.73 | 0.7959 | 100.0% |
| PLC-Spec-Syn (Gen 6) | 85.71 | 206.13 | 16.62 | 0.8277 | 67.7% |
| **PLC-Spec-Syn (Overall Weighted Avg.)** | **61.41** | **142.92** | **10.51** | **0.8310** | **78.3%** |

success on OSCAT, our significantly smaller Qwen-7B-FT achieved **31.6%**. Furthermore, utilizing PLC-Spec-Code in a **Retrieval-Augmented Generation (RAG)** setup with DeepSeek-V3.1 yielded a **70.0%** success rate, a 2.4× boost over the baseline, significantly outperforming zero-shot approaches.

## 4.4 RQ3: STRUCTURED VS. ALTERNATIVE SYNTHESIS

To answer RQ3, we conducted a detailed quantitative analysis comparing our PLC-Spec-Syn framework against five baseline methods. The evaluation focused on corpus complexity, generation trade-offs, and downstream fine-tuning performance.

**Corpus Complexity and Quality.** We first analyzed the intrinsic properties of the data generated by each method. The results, presented in Table 7, reveal a clear hierarchy. The data generated by PLC-Spec-Syn is significantly richer and more logically complex than that produced by the baselines. Notably, Traditional Augmentation (TA) shows almost no change in complexity scores, confirming its superficial nature. While methods like OSEP and MC produce specifications with high richness, their low **Code Generation Success Rates (40.3% and 55.0% respectively) reveal a critical flaw: their single-shot, overly complex instructions are less reliable and often contain ambiguities or conflicts. This makes them less conducive to the generation of valid industrial code.** In contrast, PLC-Spec-Syn achieves a superior balance of specification richness and generation reliability, which is crucial for creating a high-quality training corpus.

Our principled, guided evolution is superior to alternative data synthesis paradigms. We fine-tuned models on small, 500-sample corpora from each of the five baseline methods and our own. As shown in Table 19, **PLC-Spec-Syn outperforms all baselines**, demonstrating a more efficient training signal.

This performance gap stems from the quality of the training data itself. A detailed analysis in Appendix T reveals that while unguided methods generate syntactically complex code, it often contains logical conflicts. In contrast, our step-by-step process produces **layered and coherent logic**, which is critical for teaching models to generate reliable industrial code, especially when training data is limited.

## 4.5 CONCLUSION

This work presented PLC-Spec-Syn, a novel framework for generating the first large-scale, formally verified instruction-tuning dataset for the industrial control domain. By grounding our evolutionary process in established engineering principles and enforcing a rigorous multi-stage validation pipeline, we have produced PLC-Spec-Code, a high-fidelity corpus designed to address the critical scarcity of domain-specific data in this field. Our efforts represent a pioneering step towards enabling the application of modern large language models to high-stakes industrial automation tasks. We believe that PLC-Spec-Code will serve as a foundational benchmark, catalyzing future research into specialized code generation models and empowering the fine-tuning of LLMs for nuanced industrial applications.

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

## A    ETHICS STATEMENT

We are committed to conducting research responsibly and have adhered to the ICLR Code of Ethics throughout this work. Our study focuses on data generation and evaluation, and we have taken the following steps to address potential ethical considerations.

**Dataset and Reproducibility.**    A primary contribution of this work is the release of our new corpus, PLC-Spec-Code, which we are making publicly available to foster further research and enhance reproducibility in the field. For our downstream evaluations, we have prioritized the use of publicly accessible benchmarks. The datasets used for testing, including OSCAT, the Siemens LGF Library, and the Siemens Competition dataset, are all publicly available, with the exception of the Agents4PLC corpus, which is clearly marked as non-public. As noted in the paper, we obtained access to Agents4PLC through direct correspondence with its authors, and we state this transparently to ensure clarity on the reproducibility of our results on that specific benchmark.

**Human Subjects.**    Our methodology includes a human evaluation study to assess the quality of the generated specifications. The participants, consisting of graduate students and an industrial engineer, were fully informed about the purpose and procedure of the study and voluntarily consented to participate. The study did not involve the collection of any personally identifiable or sensitive information, and the tasks were focused solely on rating the technical quality of anonymous specifications.

## REPRODUCIBILITY STATEMENT

We are committed to ensuring the reproducibility of our research. To this end, we provide detailed descriptions of our methodology, experiments, and the full dataset generated by our work.

**Dataset Availability.**    The primary artifact of our work is the PLC-Spec-Code corpus. The complete, formally verified dataset will be progressively released to the research community to foster further research.

**Methodology and Algorithm.**    For a complete and formal description of the data generation pipeline sufficient for reimplementation, we provide detailed pseudocode in Appendix B. Furthermore, the specific engineering principles and constraints that guide the evolutionary process are detailed extensively in the appendix, including the formal definition of Smooth Evolution Pacing (SEP) in Appendix H, the multi-axis evolution rules in Appendix D, and the structured specification format in Appendix U.

**Experimental Setup and Evaluation.**    All details required to reproduce our downstream fine-tuning and evaluation results are provided. The experimental setup, including the models, baselines, and metrics, is described in Section 4. The appendix provides further specifics, including the exact evaluation datasets and zero-shot prompt template used (Appendix Q) and the complete fine-tuning protocol with hyperparameters (Appendix R). All evaluation datasets, with the exception of the non-public Agents4PLC corpus, are open-source and publicly available.

**Code Availability.**    While the source code for our data generation pipeline is not being released at this time, we have made every effort to ensure our method can be reimplemented. The detailed pseudocode in Appendix B, combined with the comprehensive description of the evolutionary rules and prompt structures, provides a complete blueprint of the algorithm's logic.

## B    THE EVO-SPEC ALGORITHM IN PSEUDOCODE

To provide a formal and comprehensive overview of our data generation pipeline, we present the end-to-end PLC-Spec-Syn framework in the form of pseudocode in Algorithm 1. This algorithm details the entire process described in our methodology, from the initial seeding and diversification phase, through the core multi-branch evolutionary loop, to the final multi-stage filtering that includes auditing, compilation, and formal verification.

---

**Algorithm 1** The Evo-Spec Data Generation Framework (Detailed)

---

1: **Input:** Seed keywords $\mathcal{K}$, Num generations $G$, Scenarios per keyword $S$
2:       Prompt templates $\mathcal{T}_{prompt}$, Branch sampling weights $\mathcal{W}_{branch}$
3:       LLM for evolution $LLM_{evolve}$, LLM for code generation $LLM_{code}$
4:       Auditor consortium $\mathcal{A}$, Audit threshold $\theta_{audit}$
5:       Compiler $Comp$, Formal verifier $Verify$
6: **Output:** Formally verified specification-code dataset $\mathcal{D}_{final}$
                                                   $\triangleright$ *Phase 1: Seeding and Diversification*
7:  $\mathcal{R}_{root} \leftarrow \emptyset$
8: **for** each $k \in \mathcal{K}$ **do**
9:     $Scenarios \leftarrow \text{Brainstorm}(k, S, LLM_{evolve})$
10:     $\mathcal{R}_{root} \leftarrow \mathcal{R}_{root} \cup \text{DeduplicateBySimilarity}(Scenarios)$
11: **end for**
                                   $\triangleright$ *Phase 2: Multi-Branch Specification Evolution*
12: $\mathcal{P}_{raw} \leftarrow \mathcal{R}_{root}$
13: **for** each $r_{root} \in \mathcal{R}_{root}$ **do**
14:     $\mathcal{P}_{gen} \leftarrow \{r_{root}\}$                          $\triangleright$ Initialize generation 0
15:     **for** $g \leftarrow 1$ to $G$ **do**
16:         $\mathcal{P}_{next\_gen} \leftarrow \emptyset$
17:         **for** each $p_{parent} \in \mathcal{P}_{gen}$ **do**
18:             $b \leftarrow \text{SampleFrom}(\mathcal{W}_{branch}[g])$ $\triangleright$ Sample branch based on stage-dependent weights
19:             $prompt \leftarrow \text{ConstructEvolvePrompt}(p_{parent}, b, \mathcal{T}_{prompt})$         $\triangleright$ Assemble the
constitutional prompt
20:             $p_{child} \leftarrow LLM_{evolve}(prompt)$
21:             $\mathcal{P}_{next\_gen} \leftarrow \mathcal{P}_{next\_gen} \cup \{p_{child}\}$
22:         **end for**
23:         $\mathcal{P}_{gen} \leftarrow \mathcal{P}_{next\_gen}$
24:         $\mathcal{P}_{raw} \leftarrow \mathcal{P}_{raw} \cup \mathcal{P}_{gen}$
25:     **end for**
26: **end for**
                              $\triangleright$ *Phase 3: Auditing, Code Generation, and Verification*
27: $\mathcal{P}_{audited} \leftarrow \emptyset$
28: **for** each $p \in \mathcal{P}_{raw}$ **do**
29:     $total\_score \leftarrow 0$
30:     **for** each $Auditor_i \in \mathcal{A}$ **do**                           $\triangleright$ Multi-agent audit
31:         $total\_score \leftarrow total\_score + Auditor_i(p)$
32:     **end for**
33:     **if** $(total\_score/|\mathcal{A}|) \geq \theta_{audit}$ **then**
34:         $\mathcal{P}_{audited} \leftarrow \mathcal{P}_{audited} \cup \{p\}$
35:     **end if**
36: **end for**
37: $\mathcal{D}_{final} \leftarrow \emptyset$
38: **for** each $p_{spec} \in \mathcal{P}_{audited}$ **do**
39:     $c_{code} \leftarrow LLM_{code}(p_{spec})$
40:     **if** $Comp(c_{code}) ==$ True **then**                   $\triangleright$ Syntactic check with matiec
41:         $\Phi_{props} \leftarrow \text{ExtractProperties}(p_{spec})$      $\triangleright$ Extract LTL/CTL properties from spec
42:         **if** $Verify(c_{code}, \Phi_{props}) ==$ True **then**       $\triangleright$ Semantic check with nuXmv
43:             $\mathcal{D}_{final} \leftarrow \mathcal{D}_{final} \cup \{(p_{spec}, c_{code})\}$
44:         **end if**
45:     **end if**
46: **end for**
47: **return** $\mathcal{D}_{final}$

---

## C  THEORETICAL GROUNDING OF EVOLUTION BRANCHES

Evo-Spec employs six orthogonal evolution branches to systematically increase task complexity. Table 8 provides a detailed summary of the theoretical basis and industrial significance for each

of these branches, demonstrating that they are grounded in established engineering practices and standards.

Table 8: Theoretical grounding and industrial importance of the six orthogonal evolution branches.

| Branch Dimension | Core Concept & Importance | Theoretical Basis & Industrial Significance |
|---|---|---|
| **Functional Extension** | Expands primary operational logic by adding I/O, states, and control rules; defines the system's core purpose and capabilities. | **Importance:** Logic structure and correctness determine whether a system performs its intended task. We adhere to **IEC 61131-3 standard** so generated logic is syntactically/semantically aligned with PLC practice (International Electrotechnical Commission, 2013a). For sequencing, we reflect finite-state models and Statecharts (Harel, 1987), consistent with PLC textbooks (Bolton, 2015; Hughes, 2011). |
| **Safety Augmentation** | Introduces safety-critical logic (e.g., emergency stops, interlocks) designed to override all other functions and force a safe state. | **Importance:** Safety is mandated in automation. We follow principles of Safety Instrumented Systems per **IEC 61508** and **ISO 13849-1** (International Electrotechnical Commission, 2010; International Organization for Standardization, 2015), adopting robust, fail-safe patterns (Goblet, 2004; Storey, 1996). |
| **Performance Optimization** | Improves efficiency, throughput, and robustness via debouncing, energy-saving modes, cycle-time control. | **Importance:** Performance impacts OEE (Nakajima, 1988). Predictable timing stems from hard real-time scheduling (Buttazzo, 2011; Liu & Layland, 1973); optimizations follow modern control principles (Ogata, 2010). |
| **Maintenance & Diagnostics** | Adds non-intrusive monitoring (cycle counters, runtime accumulators) for asset health and fault visibility. | **Importance:** Reduces downtime and supports ISO 55001-aligned asset management (International Organization for Standardization, 2014). Enables CBM/PHM practices (Jardine et al., 2006; Goebel et al., 2017). |
| **Interoperability Extension** | Connects controllers to SCADA/MES layers via interface variables and exchange schemas (Industry 4.0 context). | **Importance:** Vertical/horizontal integration is guided by **OPC UA (IEC 62541)** and **ISA-95/IEC 62264** (International Electrotechnical Commission, 2020; 2013b). In our dataset, ST programs expose abstract interface stubs; runtime protocols are modeled via metadata, avoiding implementation-specific assumptions while preserving integration intent. |
| **Contextual Complication** | Introduces adaptability to real-world dynamics (multi-recipe production, multi-machine coordination). | **Importance:** Mass customization requires agility. Grounded in Flexible Manufacturing Systems (Groover, 2014; Wadhwa & Rao, 1989), adaptive control (Astrom & Wittenmark, 2013), and multi-agent paradigms for decentralized manufacturing (Monostori et al., 2006). |

# D DETAILED EVOLUTION RULES BY BRANCH

This section details the sophisticated prompt engineering that underpins our framework. The evolution rules, summarized in Table 9, are not just high-level guidelines but are enforced by a multi-part "constitutional" prompt given to the LLM. This engineered prompt ensures every evolutionary step is constrained, consistent, and compliant with our principles. Below, we break down its key architectural components.

**Modular Prompt Architecture.** Each complete prompt is dynamically assembled from three components: a common header, a branch-specific body, and a common footer. This modularity allows us to enforce global rules consistently while providing highly specialized instructions for each of the six evolutionary branches.

**The Constitution: Common Rules and Constraints.** The common header and footer sections establish non-negotiable rules that govern every generation, regardless of the branch. Key directives include:

- **Cumulative Logic and Monotonic Complexity:** The prompt strictly enforces that the new specification must be a superset of the parent. It explicitly forbids deleting or renaming existing variables and includes a 'Counts Check' section where the LLM must verify that the number of inputs, outputs, and internal variables is non-decreasing.

- **Scenario Grounding and Naming Conventions:** To ensure industrial relevance, the prompt forces the LLM to derive all new variable names thematically from the original 'seed_keyword'. It explicitly disallows generic names (e.g., 'Input1', 'Timer1') in favor of descriptive, scenario-specific tags (e.g., 'Conveyor_Jam_Sensor', 'Tank_Agitator_TON').

- **Chain-of-Thought and Self-Validation:** Before generating the final specification, the LLM is required to output a JSON block detailing its reasoning. This includes selecting an addition, integrating it, and scoring its own adherence to principles like minimalism and safety preservation. This forces a structured, self-critical reasoning process.

**Guided Creativity: Branch-Specific Menus.** The core of each branch-specific prompt is a set of carefully curated "allowed additions," which are presented to the LLM as a limited menu of choices appropriate for the current evolutionary stage (early, mid, or late). This menu is the direct implementation of the rules summarized in Table 9. This approach guides the LLM's creativity, preventing it from making random, overly complex, or irrelevant changes. Each branch prompt also assigns the LLM a specific professional persona (e.g., "You are a Safety Engineering Specialist...") and defines a critical priority (e.g., "Safety logic SHALL override all functional logic") to ensure the enhancement is thematically sound.

**Enforcing Structure: The Unified Output Template.** Finally, the footer provides a rigid, comprehensive template for the output specification. This ensures every generated item is highly structured and machine-readable, containing sections for metadata, I/O, formal logic expressions, safety requirements, and even test scenarios for verification. This consistency is crucial for the downstream automated processing of the generated corpus, including the parsing of specifications for the formal verification pipeline.

Table 9: Detailed Evolution Rules and Examples by Branch and Stage.

| Branch Dimension | Engineering Objective | Detailed Evolution Rules (from Prompts) |
|---|---|---|
| **Functional Extension** | Expand the core operational logic and capabilities of the control system. | **Early:** Add one `BOOL` variable. Options: `System_Enable` (master switch) or `Running_Light` (status indicator). 
 **Mid:** Add one variable. Options: `Start_Delay_TON`, `Mode_Selector` (INT/CASE), or a `Manual_Override` (BOOL). 
 **Late:** Introduce one advanced ST syntax family. Options: `ENUM/CASE` for state handling, `ARRAY/FOR` for data buffering, `TOF` for advanced timing, `CTU` for counting, `R_TRIG` for edge detection, `STRUCT` for data grouping, a reusable `FUNCTION`, `STRING` manipulation, or a `WHILE` loop. |
| **Safety Augmentation** | Introduce safety-critical logic that overrides standard functions to prevent harm. | **Early:** Add one `BOOL` safety input. Options: `E_STOP_NC`, `GUARD_DOOR_NC`, or a `LIGHT_CURTAIN_NC`. 
 **Mid:** Add one safety item. Options: `SAFETY_RESET_REQ` push-button, an `SR/RS` latch for faults, `EDM_FEEDBACK` for contactor monitoring, or `DUAL_CHANNEL_CHECK` logic. 
 **Late:** Introduce one advanced safety concept. Options: a `Max_Cycle_TON` as a watchdog, a safety state manager using `ENUM/CASE`, an `ARRAY` for fault logging, a safe subroutine (`FUNCTION`), or logic for two-hand control. |
| **Performance Optimization** | Improve efficiency, throughput, and robustness without altering core functionality. | **Early:** Add descriptive metrics only (no new variables). Options: a `Response_Time_Target` or a `Debounce_Requirement`. 
 **Mid:** Implement one concrete item. Options: an `Input_Debounce_Filter` (TON) or an `Output_Settling_Timer` (TON). 
 **Late:** Implement one concrete feature. Options: a `CTU`-based throughput counter, an `ARRAY/FOR` loop for signal averaging, an `Idle_Timeout` for energy saving, a `FUNCTION` for OEE calculation, or functions for slew rate limiting or duty cycle guarding. |
| **Maintenance & Diagnostics** | Add non-intrusive monitoring features for asset health and predictive maintenance. | **Early:** Add one internal monitoring variable. Options: a `Cycle_Counter` (UINT) or a latched `First_Fault_Flag`. 
 **Mid:** Add one maintenance item. Options: a `Runtime_Accumulator` (TIME) or a `Maintenance_Due_Indicator` output. 
 **Late:** Introduce advanced diagnostics. Options: logging fault codes into an `ARRAY`, grouping all diagnostic data into a single `STRUCT`, a `FUNCTION` to calculate a wear factor or MTBF, or bitwise packing of alarms into a `WORD`. |
| **Interoperability Extension** | Integrate the control system with the wider factory ecosystem (e.g., SCADA, MES). | **Early:** Add one simple boolean interface. Options: a read-only `Remote_Status_Bit` or a `Remote_Enable_Request` input. 
 **Mid:** Add one basic communication item. Options: a 1Hz `Comms_Heartbeat` signal or a writable `Remote_Setpoint` (REAL). 
 **Late:** Add one advanced communication object. Options: a `SCADA_Interface` STRUCT, a `STRING` for a recipe ID from an MES, a `Command_Word` or `Status_Word` (WORD), or a `DINT` for a timestamp. |
| **Contextual Complication** | Make the system adaptive to real-world operational variations. | **Early:** Add one simple conditional statement. Options: an `IF/ELSE` block to handle a `Multi_Product_Threshold` or an `Environmental_Mode`. 
 **Mid:** Add one coordination input or simple ramp. Options: require an `Upstream_Process_Ready` signal or implement a `Soft_Start_Ramp` with a `TON`. 
 **Late:** Implement a complex adaptive system. Options: full recipe management (`ARRAY OF STRUCT`), adaptation to variable loads (analog input), downstream handshaking (`WHILE`), tool wear compensation (`FUNCTION`), or batch step tables. |

# E  SCENARIO BRAINSTORMING FROM SEED KEYWORDS

The first step in our data generation pipeline, preceding the creation of the Gen 0 root specifications, is a creative brainstorming phase designed to generate a diverse set of operational contexts, which

we term **Scenarios**. This step is crucial for ensuring that our dataset covers a wide range of realistic industrial challenges, moving beyond the literal interpretation of the initial seed keywords.

**Methodology.** For each of the 111 seed keywords, we leverage a large language model (LLM) to perform a creative brainstorming process. This process generates diverse and plausible control scenarios by systematically exploring variations along multiple axes, including:

- **Scale and Complexity:** From a single piece of equipment to a multi-stage system.
- **Operational Focus:** Shifting the primary goal from basic functionality to safety, efficiency, quality control, or maintenance.
- **Technology and Context:** Incorporating different sensor/actuator types and considering specific real-world industrial constraints or environmental factors.

This process is repeated to generate multiple ($S = 4$) distinct scenarios for each seed keyword, resulting in an initial pool of diverse and context-rich task ideas.

**Example.** Given the seed keyword, "An automated conveyor belt system for material transport," this brainstorming process does not simply produce a start/stop task. Instead, it might generate more sophisticated scenarios such as:

- A conveyor system with **barcode scanners** for dynamic routing of packages in a logistics center.
- A variable-speed conveyor that uses **ultrasonic sensors** to maintain a buffer between items, optimizing throughput.
- A conveyor in a food processing plant with **vision systems** for quality inspection and a pneumatic reject mechanism.

These scenarios then serve as the rich, contextual basis from which the structured Gen 0 root specifications are formulated, ensuring that the entire evolutionary lineage begins from a place of industrial relevance and diversity.

## F    REPRESENTATIVE SEED KEYWORDS AND RESULTING DOMAIN COVERAGE

As described in our methodology, the data generation process was initiated from 111 seed keywords. These keywords were sourced from a comprehensive review of canonical industrial automation textbooks, public application notes from major equipment vendors, and established online repositories of PLC programming examples. They were not chosen arbitrarily but were carefully curated to serve as robust and meaningful starting points for the evolutionary process.

**Seed Keyword Selection Criteria.** Each of the 111 keywords was selected according to three core principles to ensure relevance, clarity, and evolutionary potential:

- **Domain Relevance:** The scenario must represent a canonical task from a recognized industrial automation sector.
- **Functional Clarity:** The keyword must articulate a clear primary function (e.g., "cutting," "controlling," "dispensing") and name its core, tangible physical components.
- **Potential for Evolution:** The task must be grounded in identifiable physical systems (e.g., "conveyor belt," "heating vessel") that allow for meaningful and complex evolution across multiple engineering dimensions, such as adding safety interlocks, performance optimizations, or maintenance diagnostics.

Table 10 provides a **representative sample** of these keywords, categorized by the industrial domains they cover, to illustrate the breadth of our starting points.

To quantitatively validate that the subsequent brainstorming phase successfully expanded these seeds into a diverse set of applications, we analyzed the domain distribution of the resulting 347 classified

Table 10: A Representative Sample of Seed Keywords by Industrial Domain.

| Industrial Domain | Representative Seed Keywords |
| --- | --- |
| **Manufacturing & Assembly** | • An automated robotic welding cell for automotive assembly
• An automated semiconductor fabrication system with clean room robots
• An automated tablet pressing and coating system with real-time monitoring
• An automated aerospace composite layup system for aircraft parts |
| **Material Handling & Logistics** | • An automated conveyor belt system for material transport
• An automated palletizing system for stacking packages onto pallets
• An automated guided vehicle (AGV) network for autonomous facility logistics
• An automated sorting station using vision for parcel distribution |
| **Process Control** | • An automated chemical batching and mixing system with recipe management
• An automated wastewater treatment plant with oxygen and pH control
• An automated brewery control system for beer production
• An automated distillation column control system for separation processes |
| **Energy, Resources & Utilities** | • An automated wind turbine control system for renewable energy
• An automated boiler control system in a power plant
• An automated drilling rig control system for oil extraction |
| **Infrastructure & Agriculture** | • An automated building HVAC control system
• An automated traffic light control system with adaptive sensing
• An automated greenhouse climate control system for temperature and humidity
• An automated irrigation and fertigation system for precision agriculture |
| **Safety & Monitoring** | • An automated gas leakage detection and auto dialing system for safety monitoring
• An automated multi-channel fire alarm system for facility protection
• An automated predictive maintenance system using vibration analysis for machinery |

root scenarios. Figure 4 visualizes this distribution, confirming that the process achieved 100% coverage across all 6 major industrial domains. The chart illustrates a realistic distribution with a strong focus on core areas like Process Control (47.0 %) and Manufacturing & Assembly (17.2%), while also demonstrating comprehensive breadth by including scenarios from all other targeted sectors. This provides strong evidence for the diversity of the final PLC-Spec-Code dataset.

## G   MULTI-BRANCH EVOLUTION AND BRANCH SCHEDULING

As described in our methodology, the core of PLC-Spec-Syn is a principled, guided evolution process. This appendix provides the full text describing this strategy and the specific probabilistic weights used to guide the curriculum.

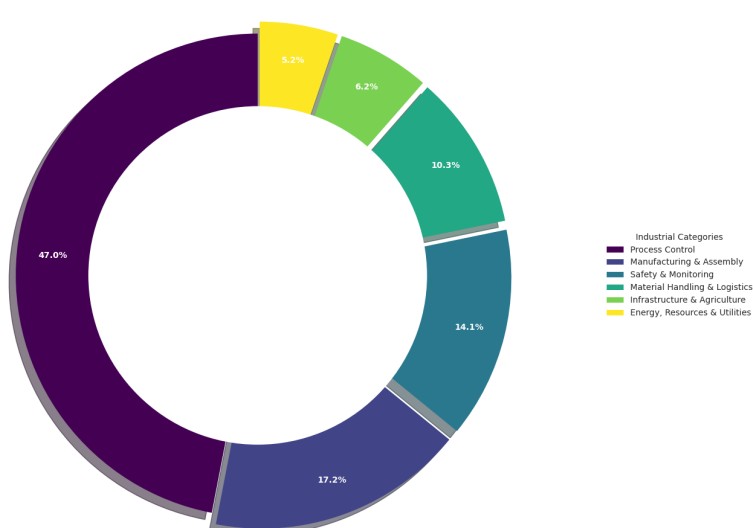

Figure 4: Distribution of 347 classified root scenarios across our 6 major industrial domains. The chart shows 100% coverage and a realistic, non-uniform distribution, confirming that the brainstorming phase successfully generated a comprehensive and diverse set of tasks representative of the industrial automation landscape.

**Multi-Branch Evolution Strategy** With the 347 root specifications established, we applied our **principled, guided** multi-axis evolution. The process uses six orthogonal branches—**functionality**, **safety**, **performance**, **maintenance**, **interoperability**, and **contextual complication**—derived from industrial standards like **IEC 61508**. To ensure pedagogically sound complexity growth, we implemented **Smooth Evolution Pacing (SEP)**, which constrains modifications to be cumulative and monotonically non-decreasing over six generations, grouped into early, mid, and late stages. At each generation, a branch was sampled using stage-dependent probabilistic weights (Table 11). This evolution was guided by branch-specific prompts enforcing a fixed output structure, cumulative logic, safety priority, scenario grounding, and self-verification.

Table 11: Stage-dependent probabilities for branch selection.

| Evolution Branch | Early (Gens 1–2) | Mid (Gens 3–4) | Late (Gens 5–6) |
|---|---|---|---|
| Functionality | 35% | 23% | 13% |
| Safety Augmentation | 35% | 19% | 14% |
| Performance Optimization | 5% | 19% | 23% |
| Maintenance & Diagnostics | 10% | 18% | 18% |
| Interoperability Extension | 5% | 11% | 17% |
| Contextual Complication | 10% | 10% | 15% |
| **Total** | **100%** | **100%** | **100%** |

## H FORMAL DEFINITION OF SMOOTH EVOLUTION PACING (SEP)

As described in our methodology, **Smooth Evolution Pacing (SEP)** is the set of constraints that governs the evolutionary process in PLC-Spec-Syn. It is designed to ensure that task complexity increases in a gradual, logical, and pedagogically sound manner, mirroring the structured way industrial projects are developed. SEP transforms the data generation from an unconstrained complexity increase into a curriculum of realistically layered engineering tasks.

**Formal Definition.** Let $S_g$ be a specification at generation $g$, and let $I(S_g)$, $O(S_g)$, and $M(S_g)$ be the sets of input, output, and internal variables in that specification, respectively. Let $T(S_g)$ be the set of timer function blocks (e.g., `TON`) used in $S_g$. An evolutionary step from a parent specification $S_g$ to a child specification $S_{g+1}$ is SEP-compliant if and only if all of the following properties hold:

1. **Property of Cumulative Logic (Superset Rule):** The functionality described by $S_{g+1}$ must be a strict superset of the functionality of $S_g$. No existing logic, variables, or requirements from $S_g$ may be removed or altered in $S_{g+1}$.

2. **Property of Monotonic Complexity (Non-decreasing Counts):** The number of variables must be monotonically non-decreasing. Formally:

$$|I(S_{g+1})| \geq |I(S_g)|, \quad |O(S_{g+1})| \geq |O(S_g)|, \quad |M(S_{g+1})| \geq |M(S_g)|$$

3. **Property of Bounded Increments (Stage-gated Growth):** The number and type of new variables added in a single generation are strictly limited by the evolutionary stage. As implemented in our prompts, these caps are:
   - **Early Stage (Gens 1-2):** The total number of new variables, $\Delta_{vars} = |I(S_{g+1}) \setminus I(S_g)| + |O(S_{g+1}) \setminus O(S_g)| + |M(S_{g+1}) \setminus M(S_g)|$, must satisfy $\Delta_{vars} \leq 1$. The new variable type must be `BOOL`. No timers may be added.
   - **Mid Stage (Gens 3-4):** The increment is typically limited to $\Delta_{vars} \leq 1$.
   - **Late Stage (Gens 5-6):** The increment is relaxed to allow for more complex additions, such as a new I/O variable plus a new internal variable ($\Delta_{vars} \leq 2$), or the introduction of advanced data structures like `ARRAY`s or `STRUCT`s.

4. **Property of Feature Cooldown (Timer Exclusivity):** The introduction of certain complex features is temporarily restricted to prevent rapid, unstable growth. The primary example is a timer cooldown:

$$T(S_g) \neq \emptyset \implies T(S_{g+1}) \setminus T(S_g) = \emptyset$$

This means if a timer was added in generation $g$, no new timer may be added in generation $g+1$. This rule encourages the system to integrate and stabilize one complex feature before adding another.

**On Using Variable Count as a Complexity Proxy.** A core tenet of SEP is the use of a non-decreasing variable count as a primary proxy for complexity. This choice is deliberate and grounded in both practical and theoretical considerations:

- **Rationale and Importance:** In industrial control systems, the number of I/O points and internal state variables directly corresponds to the scope and logical intricacy of the task. Unlike more abstract measures of difficulty, the variable count is a simple, objective, and automatically verifiable metric. By enforcing a monotonic increase in this count, we prevent the model from simplifying or fundamentally altering the task in later generations, ensuring a traceable and consistently expanding logical footprint. This constraint is crucial for creating a coherent curriculum where complexity is built layer by layer.

- **Grounding in Engineering Practice:** This principle directly mirrors real-world engineering workflows. Industrial control systems are rarely replaced; they are augmented. When a new feature is required—such as an additional sensor, a new safety interlock, or a maintenance alarm—engineers add new tags (variables) to the existing program. The SEP rule of non-decreasing variables simulates this additive and iterative process of brownfield development, where new functionality must be integrated without compromising the existing, validated core. It enforces a form of backward compatibility at the specification level, a critical practice in industrial engineering.

**Justification and Significance of SEP.** These properties are not arbitrary; they are derived from established principles of software and control systems engineering.

- The **Cumulative Logic** and **Monotonic Complexity** properties ensure that each new task is a logical and traceable extension of its parent. This creates a coherent learning path for a model, teaching it how to layer new features (e.g., maintenance logic) onto an existing functional core without breaking it.

- The **Bounded Increments** property functions as a curriculum, starting with foundational concepts before moving to more complex ones. This prevents the generation of tasks that are "hard" but nonsensical, a common failure mode of unguided complexity-growth methods like Evol-Instruct.

- The **Feature Cooldown** property promotes robust integration. By forcing the system to explore other evolutionary branches after introducing a significant feature like a timer, it generates more well-rounded and realistic tasks where different engineering aspects (e.g., performance and safety) must interact.

Together, the SEP rules ensure that the PLC-Spec-Code dataset represents a collection of not just complex tasks, but of logically coherent and realistically developed systems, providing a superior training signal for downstream models.

## I    ILLUSTRATION OF THE BRANCHING EVOLUTION PROCESS

Figure 5 provides a conceptual sketch of the multi-branch evolution strategy. Starting from a single keyword, multiple scenarios are generated. Each of these becomes a root specification, which then serves as the starting point for a six-generation evolution tree. At each generation, a branch (e.g., functionality, safety) is stochastically selected to incrementally add a new, stage-appropriate feature, resulting in a diverse and complex set of control tasks.

## J    FORMAL VERIFICATION PIPELINE FOR SEMANTIC EQUIVALENCE

This appendix details the process for formally verifying the semantic equivalence between our structured specifications and the generated Structured Text (ST) code. Our verification pipeline is inspired by the framework in **LLM4PLC** (Fakih et al., 2024), leveraging the `nuXmv` symbolic model checker (Cavada et al., 2014). We demonstrate how the rich, semi-formal nature of our specifications enables a highly robust and reproducible verification workflow.

### J.1    VERIFICATION FRAMEWORK: PLANT, CONTROLLER, AND SPECIFICATION

As described in LLM4PLC, formal verification in our context is not merely a check on the code itself, but an analysis of the interaction between the controller and its environment. Our verification framework therefore comprises three core components modeled in the SMV language for `nuXmv`:

**1. The Plant Model.**   This component formalizes the physical behavior of the industrial process. It is derived directly from the `I/O Requirements` and control objective sections of our specification, modeling how actuators affect sensors (e.g., opening an 'InletValve' causes 'TankLevel' to rise) and the environment's inherent rules.

**2. The Controller Model.**   This is the ST code generated by our LLM pipeline, translated into an equivalent SMV state machine representation.

**3. The Specification Properties.**   This component contains the formal properties the controller must satisfy. Crucially, thanks to our structured specification format, these properties are derived through a highly deterministic process, as detailed below.

The goal of verification is to have `nuXmv` prove that for all possible behaviors of the integrated **Plant-Controller system**, all defined **Specification Properties** hold true.

### J.2    DERIVING FORMAL PROPERTIES FROM STRUCTURED SPECIFICATIONS

Our detailed specification format allows for a more direct and reliable translation to temporal logic than parsing free-form text. We use a multi-pronged approach to extract formal properties:

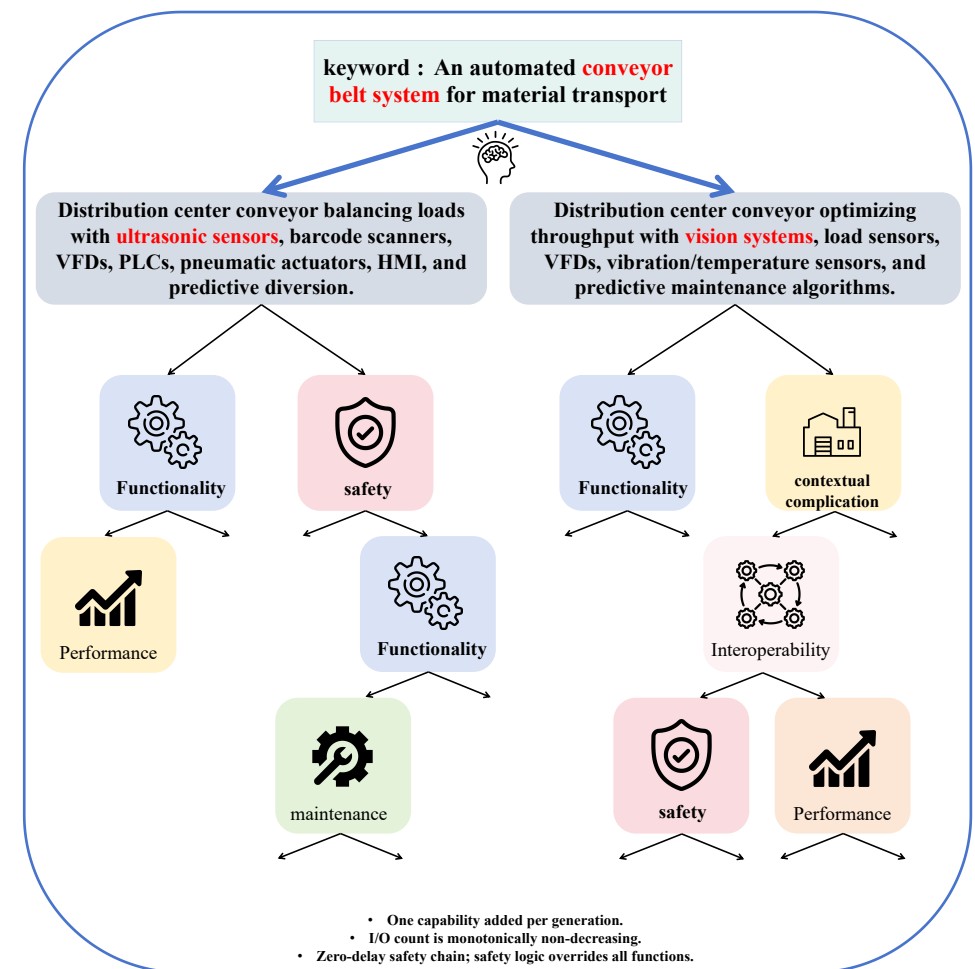

Figure 5: Illustrative branching evolution. A single *keyword* yields multiple *scenarios*; each scenario becomes a *root specification* that evolves along stochastically chosen branches, generating a spectrum of realistic control tasks with increasing complexity.

**Direct Translation from Formal Logic Expressions.** Our specifications include a `Formal Logic Expression` section containing boolean equations (e.g., `MAINTENANCE_ALERT := A AND (B OR C)`). A deterministic Python script parses these expressions and directly translates them into equivalent LTL/CTL properties that check for the correct assignment under all conditions. This is our primary and most reliable method for formalizing the core control logic.

**Rule-Based Instantiation from Verification Scenarios.** The `Verification Requirements` section provides explicit test scenarios in a "trigger-response" format (e.g., "`Watchdog Timeout` → `PUNCH_SAFETY_PERMISSIVE = FALSE`"). These scenarios are systematically converted into formal properties. For instance, the watchdog timeout scenario directly maps to the LTL formula `G((cycle_timer > PT) -> !PUNCH_SAFETY_PERMISSIVE)`, providing a precise formalization of critical safety and operational requirements.

**LLM-Assisted Template Matching for Prose Requirements.** For high-level requirements described only in prose (e.g., in the `Safety Requirements` section), we employ an LLM-based template-matching approach, similar to LLM4PLC. This method is used to capture overarching principles not expressed as explicit equations.

## K    METHODOLOGY FOR QUANTITATIVE COMPLEXITY AND SIMILARITY ANALYSIS

To quantitatively assess our dataset, we developed a static analysis program that evaluates each specification-code pair across three distinct aspects: the richness of the natural language specification, the structural complexity of the generated code, and the semantic similarity between them.

### K.1    SPECIFICATION RICHNESS ANALYSIS

The richness of each natural language specification ('input') is quantified by parsing the text and measuring the density of its technical requirements. This is based on two primary metrics: **I/O Complexity** (counting declared variables) and **Conceptual Density** (counting keywords from a curated dictionary representing core industrial concepts). The final `richness_score` is a weighted sum of these counts.

### K.2    CODE COMPLEXITY ANALYSIS

The complexity of the generated ST code ('output') is evaluated using a combination of standard software metrics: **Physical Metrics** (Lines of Code, variable count), **Syntactic Diversity** (variety of IEC 61131-3 keywords used), and **Logical Complexity**. The latter is estimated using **Cyclomatic Complexity**, a classic metric calculated by counting all control flow statements and logical operators to quantify the number of independent paths through the program.

### K.3    SEMANTIC SIMILARITY ANALYSIS

To measure semantic alignment, we employ a pre-trained **Sentence Transformer** model (`all-MiniLM-L6-v2`) to convert the specification and code into dense vector embeddings. We then calculate the **Cosine Similarity** between these vectors, where a score of 1 signifies a perfect semantic match.

### K.4    VISUALIZING THE METRICS ACROSS GENERATIONS

To illustrate the practical results of applying these analysis metrics to our generated corpus, the following figures show the trends across all seven evolutionary generations. Figure 6 shows the steady increase in the average scores for all three complexity metrics, while Figure 7 provides a more detailed view of the growth in logical complexity and its variance.

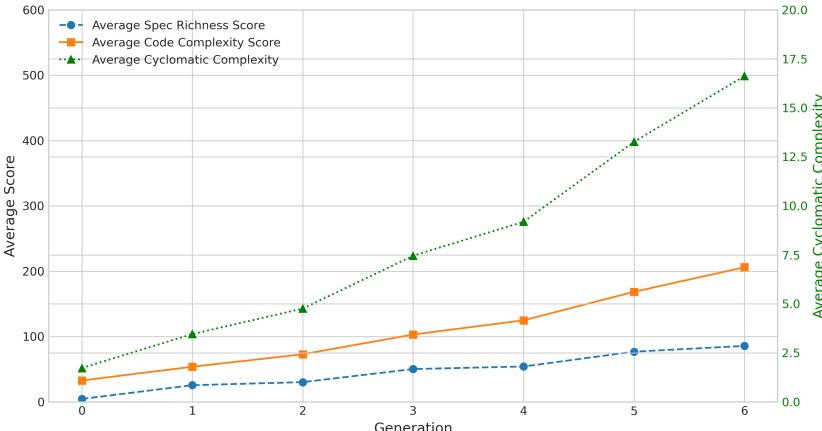

Figure 6: Trend lines for average Specification Richness, Code Complexity, and Cyclomatic Complexity across generations.

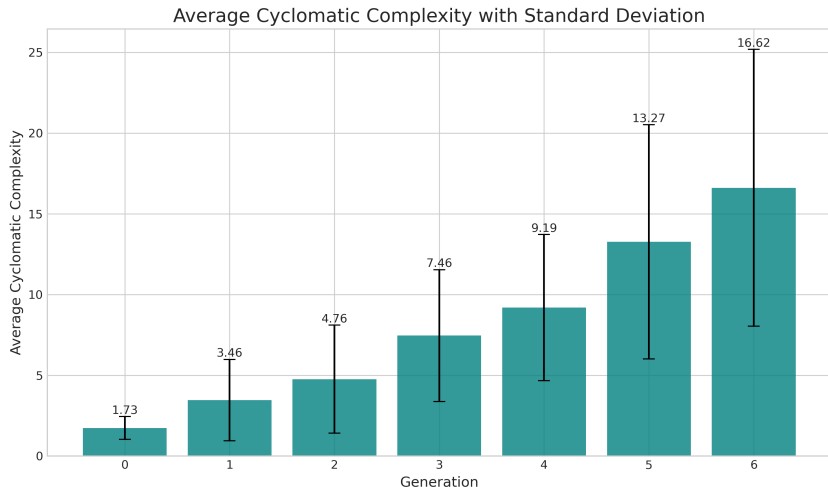

Figure 7: Bar chart of Average Cyclomatic Complexity. The height of each bar is the mean, and the error bar represents the standard deviation, showing an increase in both the average complexity and its variance in later generations.

## L    DETAILED GENERATION AND VALIDATION FUNNEL STATISTICS

This section provides the detailed statistics for our end-to-end data generation and validation pipeline. The process begins with a large pool of raw, LLM-generated specifications and applies a series of increasingly stringent filters: a multi-agent quality audit, compilation for syntactic correctness, and finally, formal verification for semantic equivalence using `nuXmv`.

Table 12 breaks down the number of surviving specification-code pairs at each stage, aggregated by their evolutionary generation. The data illustrates our rigorous filtering process and highlights the progressive difficulty of generating valid and logically correct code in later generations.

Table 12: End-to-end generation and validation funnel statistics. The table shows the complete filtering from raw specifications to the final set of formally verified specification-code pairs.

| Generation | Raw Specs (Initial Pool) | Audited Specs | Compiling Code | Verified Pairs (nuXmv) | Audit Pass Rate (%) | Compile Success Rate (%) | Overall Success Rate (%) |
|---|---|---|---|---|---|---|---|
| Gen 0 | 347 | 347 | 347 | 344 | 100.0 | 100.0 | 99.1 |
| Gen 1 | 689 | 682 | 668 | 641 | 99.0 | 97.9 | 93.0 |
| Gen 2 | 1,335 | 1,295 | 1,230 | 1,119 | 97.0 | 95.0 | 83.8 |
| Gen 3 | 2,293 | 2,178 | 1,960 | 1,696 | 95.0 | 90.0 | 74.0 |
| Gen 4 | 3,545 | 3,261 | 2,674 | 2,208 | 92.0 | 82.0 | 62.3 |
| Gen 5 | 5,128 | 4,513 | 3,370 | 2,649 | 88.0 | 74.7 | 51.7 |
| Gen 6 | 7,025 | 5,971 | 4,043 | 3,012 | 85.0 | 67.7 | 42.9 |
| **Total** | **20,362** | **18,247** | **14,292** | **11,669** | **89.6** | **78.3** | **57.3** |

**Analysis of the Generation Funnel Statistics**    The data presented in Table 12 provides a quantitative validation of our multi-stage filtering pipeline and offers key insights into the challenges of generating complex, high-quality industrial control code. The trends across the evolutionary generations directly reflect the curriculum-based complexity introduced by the PLC-Spec-Syn methodology.

- **Controlled Decline in Audit Pass Rate:** The `Audit Pass Rate` shows a steady, controlled decline from 100% (Gen 0) to 85.0% (Gen 6). This trend validates our multi-agent audit, which effectively catches the increasing ambiguity and potential inconsistencies inherent in more complex, feature-rich specifications.

- **Significant Drop in Compile Success Rate:** The steep decline in the `Compile Success Rate` to 67.7% for late-stage generations starkly illustrates a core challenge: modern LLMs

struggle to maintain syntactic correctness when generating code for complex, multi-faceted requirements. This bottleneck underscores the critical need for a high-quality, diverse dataset like PLC-Spec-Code.

- **Impact of Formal Verification:** The final filtering step, formal verification, reveals a crucial and realistic trend: the likelihood of a syntactically correct program containing a logical flaw increases with its complexity. The verification pass rate declined from a near-perfect **99.1%** for the simple Gen 0 programs to just **74.0%** for the highly complex Gen 6 programs. This proves that compilation alone is insufficient for guaranteeing correctness and that as complexity grows, rigorous semantic checks like `nuXmv` become exponentially more critical for catching subtle errors in state transitions or safety logic.

- **The Value of Rigorous End-to-End Filtering:** The final `Overall Success Rate`, which reflects the cumulative effect of all three filters, is just **57.3%**. For the most complex tasks in Gen 6, this rate falls dramatically to **42.9%**. This means that for every 10 raw complex specifications we started with, just over 4 survived our entire quality pipeline. This finding is a powerful testament to the rigor of our process and highlights the value and rarity of the final, formally verified pairs in our dataset.

In summary, the generation funnel statistics quantitatively justify our methodology. The progressive difficulty shown in the pass rates validates that PLC-Spec-Syn successfully creates a curriculum of increasing complexity, and the significant attrition rate at each stage proves the necessity of our multi-stage validation pipeline—especially formal verification—for curating a high-fidelity corpus.

# M   ST Syntax Taxonomy

Our quantitative analysis of syntactic diversity is based on a comprehensive taxonomy of 89 key tokens from the IEC 61131-3 Structured Text standard, as implemented in our evaluation script. This taxonomy, detailed in Table 13, provides a robust framework for measuring the progressive introduction of language features throughout the evolutionary process.

Table 13: The comprehensive 89-token ST syntax taxonomy used for coverage analysis.

| Feature Family | Syntactic Tokens | Count |
|---|---|---|
| POU/Blocks | PROGRAM, VAR_INPUT, VAR_OUTPUT, VAR, END_VAR, END_PROGRAM, FUNCTION, END_FUNCTION, FUNCTION_BLOCK, END_FUNCTION_BLOCK, TYPE, END_TYPE | 12 |
| DataTypes:Elementary | BOOL, INT, UINT, DINT, REAL, STRING, TIME, BYTE, WORD, DWORD, LWORD | 11 |
| DataTypes:Composite | ARRAY, STRUCT, ENUM | 3 |
| ControlFlow | IF, THEN, ELSE, ELSIF, END_IF, CASE, OF, END_CASE, FOR, TO, BY, DO, END_FOR, WHILE, END_WHILE, REPEAT, UNTIL, END_REPEAT, RETURN, EXIT | 21 |
| FunctionBlocks:Std | TON, TOF, TP, SR, RS, CTU, CTD, R_TRIG, F_TRIG | 9 |
| Operators:Boolean | AND, OR, NOT, XOR | 4 |
| Operators:Rel/Arith | :=, =, <>, >, <, >=, <=, +, −, *, /, MOD, **, .. | 14 |
| Operators:BitwiseShift | SHL, SHR | 2 |
| Literals/Consts | TRUE, FALSE, T#, TOD#, D#, DT#, 2#, 8#, 16# | 9 |
| Builtins/Conv | BOOL_TO_WORD, WORD_TO_BOOL, INT_TO_REAL, REAL_TO_INT | 4 |
| **Total Tokens** | | **89** |

# N   Detailed Analysis of Syntax Coverage Progression

As summarized in the main paper, the data in Table 3 provides strong validation for our staged evolution approach. Here, we provide a more detailed, stage-by-stage interpretation of these results.

- **Early Stage (Gen 0-2):** The process begins with foundational syntax. Gen 0 uses only the most basic block structures, boolean types, and assignments. By the end of the early stage (Gen 2), the corpus has significantly expanded to include simple IF/THEN logic and the most common standard function blocks (e.g., R_TRIG, TON), forming a solid basis for more complex logic.

- **Mid Stage (Gen 3-4):** This stage focuses on functional expansion, which is reflected in the introduction of more advanced ControlFlow constructs like CASE statements and

a broader use of `FunctionBlocks:Std`. By `Gen 4`, the coverage of operators and control flow has nearly doubled from the early stage, indicating more complex logical expressions.

- **Late Stage (Gen 5-6):** The final stages introduce the most advanced language features. We see the first appearance of `DataTypes:Composite` (e.g., `ARRAY`), the activation of almost the full range of control statements and operators, and specialized features like type conversion functions.

This progressive introduction of syntactic features provides strong quantitative evidence that Evo-Spec functions as an effective curriculum learning strategy, systematically increasing task complexity from simple control loops to feature-rich and syntactically diverse industrial programs.

## O    METHODOLOGY FOR LLM-BASED SPECIFICATION QUALITY AUDIT

All raw specifications underwent a context-aware, multi-agent audit to ensure their quality and suitability before code generation. This appendix details the methodology for that audit.

**Auditor Consortium Personas.**    The process used a committee of three diverse LLMs, each adopting a distinct professional persona to provide a holistic evaluation:

- A **Technical PLC Expert**, focusing on the technical soundness and adherence to IEC 61131-3 standards.
- A **Safety Compliance Auditor**, focusing on the correct specification of safety interlocks, fault states, and procedures.
- A **Systems Integration Engineer**, focusing on the practical implementability, clarity for technicians, and real-world relevance.

**Evaluation Rubric and Scoring.**    Each specialist agent rated every specification on a 1-5 Likert scale against the four core criteria below, interpreted through their unique professional lens. A specification was approved only if its context-aware average score across all three evaluators was $\geq 4.0$.

- **Industrial Relevance:** The plausibility of the task in a real-world industrial setting.
- **Clarity & Unambiguity:** The precision of the language, ensuring the task is well-defined.
- **Complexity Appropriateness:** Whether the specification's complexity is appropriate for its stated generation number.
- **Constraint Adherence & Preservation:** A critical check that the specification fully preserves all parent constraints and that any newly introduced logic is well-defined.

**Per-Generation Audit Results.**    Table 14 presents the average scores from this multi-persona audit for a representative sample of specifications at each evolutionary stage, validating the quality of specifications across the curriculum.

Table 14: Average scores from the multi-persona LLM audit of specification quality (1-5 scale). The consistently high scores validate the quality of specifications across all evolutionary stages.

| **Generation** | Industrial Relevance | Clarity & Unambiguity | Complexity Appropriateness | Constraint Adherence | Overall Average |
|---|---|---|---|---|---|
| Gen 0 (Base) | 4.72 | 4.91 | 4.83 | 4.98 | **4.86** |
| Gen 1 (Early) | 4.81 | 4.84 | 4.73 | 4.92 | **4.83** |
| Gen 2 (Early) | 4.83 | 4.75 | 4.71 | 4.90 | **4.80** |
| Gen 3 (Mid) | 4.92 | 4.70 | 4.81 | 4.84 | **4.82** |
| Gen 4 (Mid) | 4.89 | 4.63 | 4.78 | 4.81 | **4.78** |
| Gen 5 (Late) | 4.91 | 4.52 | 4.74 | 4.76 | **4.73** |
| Gen 6 (Late) | 4.88 | 4.45 | 4.71 | 4.73 | **4.69** |

**Analysis and Insights from Specification Audit**   The scoring trends in Table 14 offer valuable insights. The near-perfect scores for **Constraint Adherence** validate the core mechanism of our guided evolution, confirming that the process reliably builds upon parent specifications without losing information. Similarly, the consistently high scores for **Industrial Relevance** and **Complexity Appropriateness** indicate that the evolution remains grounded and well-paced.

The most insightful trend is the slight but noticeable decline in the **Clarity & Unambiguity** score in late-stage generations (from 4.91 to 4.45). This does not represent a degradation in quality, but rather reflects the inherent difficulty of articulating the complex interplay between multiple, orthogonally-evolved features (e.g., safety, performance, and maintenance logic) in a single, perfectly unambiguous natural language document. This finding suggests a potential upper bound on descriptive clarity for highly complex control tasks and highlights a key conclusion: for such tasks, the generated code becomes a **necessary formal representation**, complementing the natural language specification to provide a complete and executable definition of the requirements.

## P  HUMAN EVALUATION OF SPECIFICATION QUALITY

To validate the quality of the specifications generated by Evo-Spec, we conducted a user study with domain-knowledgeable participants.

**Participants and Procedure.**   We recruited an evaluation panel consisting of five graduate students (2 MSc, 3 PhD) specializing in industrial automation and one industrial engineer with five years of professional experience. The study consisted of two parts. In the first part, participants were shown a randomized subset of 1,000 specifications sampled from across all seven generations (gen0-gen6) and were asked to rate each on a 1-5 Likert scale according to three core quality criteria: **Logical Completeness & Correctness**, **Industrial Practicality & Safety**, and **Descriptive Clarity & Normativeness**.

In the second part, participants were shown 30 complete evolutionary lineages (from Gen 0 to Gen 6) and were asked to provide a single score for **Evolutionary Effectiveness**, judging the entire chain for its realism and logical coherence.

**Results**   The aggregated scores, averaged across all five participants, are presented in Table 15. The overall score for Evolutionary Effectiveness across the 30 lineages was **4.83**.

Table 15: Human evaluation results for specification quality, averaged across 5 domain-expert participants (1-5 scale). Scores are grouped by evolutionary stage.

| **Evolutionary Stage** | Logical Completeness | Industrial Practicality | Descriptive Clarity | Overall Quality Average |
|---|---|---|---|---|
| Gen 0 (Base) | 4.88 | 4.65 | 4.92 | **4.82** |
| Gen 1-2 (Early) | 4.85 | 4.78 | 4.81 | **4.81** |
| Gen 3-4 (Mid) | 4.79 | 4.86 | 4.65 | **4.77** |
| Gen 5-6 (Late) | 4.72 | 4.91 | 4.48 | **4.70** |

**Interpretation of Results**   The human evaluation results strongly support our claims. The consistently high scores (all averages ¿4.4) across all generations confirm the overall quality of the dataset. We note two particularly insightful trends:

- The score for **Industrial Practicality** steadily increases, peaking in the late-stage generations. This suggests that as more features are added, the tasks become more representative of real-world, multi-faceted engineering problems.

- The score for **Descriptive Clarity** shows a slight decline in the late stages. Participants noted that while the specifications were still clear, the sheer density of information in highly

evolved tasks made them more challenging to parse than simpler ones. This reflects the inherent trade-off between complexity and descriptive simplicity in technical documentation.

The outstanding score for Evolutionary Effectiveness (4.83/5.0) provides the strongest evidence that our Evo-Spec framework generates tasks that evolve in a manner that domain experts find logical and realistic.

## Q  DOWNSTREAM EVALUATION DETAILS

This section provides further details on the datasets and the prompt template used for the downstream code generation experiments.

**Evaluation Datasets**    To ensure a rigorous and comparable evaluation, our test suite is composed of four challenging industrial control datasets. Notably, three of these benchmarks—OSCAT, LGF, and the Siemens Competition set—were recently employed to validate the state-of-the-art AutoPLC framework (Yang et al., 2024), establishing them as authoritative testbeds for this domain. The datasets were used exclusively for testing and were not seen during fine-tuning.

- **CODESYS OSCAT Library (718 cases)** (oscat.de, 2024-11-03): This extensive, community-driven open-source project provides a vast collection of function blocks. Its scope is broad, covering everything from fundamental logic utilities to building automation, making it an excellent benchmark for general ST language proficiency.

- **Siemens LGF Library (151 cases)** (Siemens, 2024): The Library of General Functions (LGF) is an official, industrial-grade resource provided by Siemens. It contains highly optimized and commonly used functions, presenting a strong test for generating code that adheres to specific vendor conventions and best practices.

- **Siemens Competition Dataset (45 cases)** (biendata.xyz, 2024): Sourced from official Siemens programming competitions, this dataset contains tasks that model complete, real-world process control challenges. The problems are notable for their practical constraints and complexity, requiring multi-step reasoning beyond simple function generation.

- **Agents4PLC Corpus (187 cases)** (Liu et al., 2024): This corpus contains control problems specifically curated to test the reasoning capabilities of LLMs in PLC contexts. As this dataset is not publicly available and was obtained through direct correspondence with the authors, our main conclusions are drawn from the three public benchmarks to ensure full reproducibility. We include results on Agents4PLC to offer additional insights into model performance on tasks specifically designed for LLM reasoning.

**Zero-shot Prompt Template**    To rigorously assess the models' intrinsic code generation capabilities and the direct impact of fine-tuning, we intentionally used a minimal, zero-shot prompt for all evaluations across all models. This deliberately simple prompt was prepended to every task specification:

```
Please strictly output the complete IEC 61131-3 ST code.
```

This prompt provides no examples, hints, or chain-of-thought guidance. It forces the models to rely solely on their pre-existing knowledge (either from original pre-training or our subsequent fine-tuning). This stringent, zero-shot approach ensures that any observed performance gains in the fine-tuned models are directly attributable to the quality and diversity of the PLC-Spec-Code training data, not to sophisticated prompt engineering.

## R  DOWNSTREAM TASK EVALUATION: DETAILED RESULTS

This appendix provides the detailed methodology and full results for the downstream fine-tuning experiments summarized in subsection 4.3.

**Fine-tuning Protocol** We fine-tuned the three base models for 3 epochs using the Llama-Factory framework on the 11,669 high-quality pairs of our `PLC-Spec-Code` corpus. We used the AdamW optimizer with a learning rate of 1e-5 and a cosine learning rate schedule. All experiments were conducted on a system with two NVIDIA A100 (80GB) GPUs.

**Evaluation Protocol** For each task in the four test sets, we provided the natural language objective to the model using a zero-shot prompt. The models were tasked with generating the complete ST program. The generated code was then compiled using the `matiec` compiler. A program was marked as a "success" only if it compiled with zero errors.

**Full Results** The complete results are detailed in Table 16. The data shows that models fine-tuned on `PLC-Spec-Code` (denoted by FT) achieve substantial performance gains across all datasets. The results also provide a comprehensive comparison against several strong open-source and proprietary baseline models evaluated in a zero-shot setting. The fine-tuned `Qwen-7B-FT` model achieves the best overall performance, demonstrating the powerful combination of a strong base model and our high-quality, domain-specific training data.

Table 16: Full results of the downstream code generation task. Each cell shows the number of successful compilations and the success rate (%). FT denotes models fine-tuned on our `PLC-Spec-Code` corpus. *Non-public benchmark, results provided for additional context.*

| | Qwen-1.5B-Instruct | | Qwen-7B-Instruct | | Llama-1B-Instruct | | Open & Proprietary Baselines (Zero-shot) | | | | |
|---|---|---|---|---|---|---|---|---|---|---|---|
| Test Set (Size) | Base | FT | Base | FT | Base | FT | CodeLlama-34B | Qwen2.5-32B | DeepSeek-V3.1 | GLM-4.5 | Kimi-K2 |
| OSCAT (718) | 91 (12.7%) | **216 (30.1%)** | 108 (15.0%) | **227 (31.6%)** | 16 (2.2%) | 122 (17.0%) | 246 (34.3%) | 188 (26.2%) | 155 (21.6%) | 167 (23.3%) | 179 (24.9%) |
| Competition (45) | 6 (13.3%) | 12 (26.6%) | 8 (17.8%) | **15 (33.3%)** | 0 (0.0%) | 5 (11.1%) | 10 (22.2%) | 10 (22.2%) | 11 (24.4%) | 13 (28.9%) | 12 (26.7%) |
| LGF (151) | 7 (4.6%) | **37 (24.5%)** | 20 (13.2%) | **38 (25.2%)** | 3 (2.0%) | 28 (18.5%) | 28 (18.5%) | 25 (16.6%) | 45 (29.8%) | 21 (13.9%) | 10 (6.6%) |
| Agents4PLC (187)* | 81 (43.3%) | 114 (61.0%) | 96 (51.3%) | **143 (76.5%)** | 5 (2.7%) | 50 (26.7%) | 136 (72.7%) | 133 (71.1%) | 142 (75.9%) | 91 (48.7%) | 95 (50.8%) |

## R.1 MANUAL EVALUATION OF FUNCTIONAL CORRECTNESS

**Methodology.** While compilation success measures syntactic validity, it does not guarantee semantic correctness. To conduct a fair and direct comparison of logical reasoning, we first identified the complete set of tasks successfully compiled by both the base `Qwen2.5-Coder-1.5B-Instruct` model and our fine-tuned version. This yielded a total of 32 tasks for manual inspection. Each of these 32 pairs of outputs was then inspected by a human expert to evaluate its `semantic correctness`—that is, whether the code's logic was a plausible and correct implementation of the specification.

**Results and Analysis.** The evaluation revealed a stark difference in the models' ability to generate logically sound code.

- **Fine-tuned Model:** Our fine-tuned model produced semantically correct logic for an overwhelming **25 out of the 32 tasks (78.1%)**. Furthermore, the majority of its correct outputs were well-structured and included helpful comments that clarified the code's intent—a hallmark of high-quality, human-like programming.

- **Base Model:** In contrast, the base model produced correct logic for only **12 of the 32 tasks (37.5%)**. Its most common failure mode was generating syntactically valid but "semantically nonsensical" code. This qualitative difference is vividly illustrated by the `DRIVER_4` task, for which the code generated by both models is shown side-by-side in Figure 8.

As the figure demonstrates, the base model, when faced with a somewhat ambiguous specification, generated a convoluted program with dozens of unnecessary variables and a complex web of logic that bears no resemblance to a typical driver function. Critically, it even attempts to assign values to its own inputs (`SET` and `RST`), a fundamental logical error. Conversely, our fine-tuned model generated a concise, plausible, and commented implementation. It correctly interpreted `SET` as a master enable and `RST` as a high-priority reset, producing clean and understandable code that represents a valid solution.

**Conclusion** This direct comparison on a shared set of compiling tasks strongly corroborates our quantitative findings. Fine-tuning with the `PLC-Spec-Code` corpus does more than just

Figure 8: Side-by-side comparison for the DRIVER_4 task. The base model produces syntactically valid but illogical and over-engineered code, while the fine-tuned model generates a clean, commented, and semantically correct solution.

**Base Model Output**

```
PROGRAM DRIVER_4
VAR_INPUT
    Toggle_Mode: BOOL;
    Timeout: TIME;
    SET: BOOL;
    IN0: BOOL; IN1: BOOL;
    IN2: BOOL; IN3: BOOL;
    RST: BOOL;
END_VAR
VAR_OUTPUT
    Q0: BOOL; Q1: BOOL;
    Q2: BOOL; Q3: BOOL;
END_VAR
VAR
    IN0_1: BOOL; IN0_2: BOOL; ...
    ... (22 more VAR declarations)
    Q3_3: BOOL;
END_VAR
(* Program logic *)
IN0_1 := Toggle_Mode AND IN0;
IN0_2 := Toggle_Mode AND IN0_1;
... (10 more nonsensical assignments)
Q0_1 := IN0_1 OR IN1_1 OR ...
... (11 more nonsensical assignments)
SET := Q0_1 OR Q0_2 OR Q0_3; (* Error! *)
RST := Q1_1 OR Q1_2 OR ...;  (* Error! *)
END_PROGRAM
```

**Fine-tuned Model Output (Ours)**

```
PROGRAM DRIVER_4
VAR_INPUT
    Toggle_Mode : BOOL; (* Mode input *)
    Timeout : TIME; (* Timeout for drivers *)
    SET : BOOL;       (* Master enable switch *)
    IN0 : BOOL;       (* First input signal *)
    IN1 : BOOL;       (* Second input signal *)
    IN2 : BOOL;       (* Third input signal *)
    IN3 : BOOL;       (* Fourth input signal *)
    RST : BOOL;       (* Manual reset button *)
END_VAR
VAR_OUTPUT
    Q0 : BOOL;        (* First driver output *)
    Q1 : BOOL;        (* Second driver output *)
    Q2 : BOOL;        (* Third driver output *)
    Q3 : BOOL;        (* Fourth driver output *)
END_VAR

(* Main operational logic *)
Q0 := IN0 AND SET;
Q1 := IN1 AND SET;
Q2 := IN2 AND SET;
Q3 := IN3 AND SET;

(* Safety and reset logic *)
IF RST THEN
    Q0 := FALSE;
    Q1 := FALSE;
    Q2 := FALSE;
    Q3 := FALSE;
END_IF;
END_PROGRAM
```

improve syntax; it fundamentally enhances the model's ability to reason about industrial logic and translate specifications into semantically correct and well-structured programs. The qualitative difference—from illogical, over-engineered code to clean, commented, and correct solutions—underscores the value of our structured, curriculum-driven data generation process for instilling robust programming capabilities.

## S  DETAILED BASELINE METHODOLOGIES

This section provides a detailed description of the five baseline data generation strategies used for comparison in our experiments. Each method was designed to isolate a specific aspect of data synthesis and was used to generate a 500-sample corpus for a fair, head-to-head comparison with our PLC-Spec-Syn framework.

**Traditional Augmentation (TA-Corpus)** This baseline is designed to represent a common, low-effort data augmentation approach. Its goal is to test whether simple, non-semantic syntactic variety is sufficient to improve model performance.

- **Process:** We began by randomly sampling 500 specification-code pairs from a stratified subset of our own generated corpus. For each of the 500 programs, we used an LLM to apply **one of three** randomly selected, syntax-preserving transformations: semantic-equivalent variable renaming, converting a FOR loop to an equivalent WHILE loop, or

reordering independent logical statements. The original specification was then paired with the newly augmented code.

- **Purpose:** This method provides a baseline for the performance gains achievable through superficial syntactic changes on an already diverse and complex dataset. It helps to isolate the value of generating true semantic and structural novelty from simply rephrasing existing code structures.

**Self-Instruct Proxy (SI-Proxy)**    This baseline simulates the classic Self-Instruct paradigm, which excels at generating a wide **breadth** of new, distinct tasks from a small set of seeds. Recognizing that Self-Instruct typically prioritizes variety over complexity, we adapted the method to introduce a controlled amount of logical **depth**.

- **Process:** Starting from our 347 root specifications, we prompted the LLM to "generate a new and distinct industrial task inspired by the seed." To introduce depth, we augmented this prompt with a secondary instruction: "The new task should also incorporate one additional concept from the following list: [a fault condition, a manual override, a status indicator light]." This "depth-aware sampling" encourages the creation of novel tasks that are slightly more complex than the original seeds, while still emphasizing breadth. A pool of new specifications and their corresponding code was generated, from which we sampled 500 pairs.

- **Purpose:** This method tests the effectiveness of generating a large variety of moderately simple tasks, contrasting with PLC-Spec-Syn's strategy of evolving a smaller set of tasks to a high degree of complexity.

**Evol-Instruct Proxy (EI-Proxy)**    This baseline is designed to simulate the core principle of Evol-Instruct: increasing complexity in an unguided, generic manner.

- **Process:** We provided our 347 root specifications to the LLM with a simple, open-ended prompt, such as "Rewrite the following specification to be more complex and challenging." Unlike our PLC-Spec-Syn framework, this prompt provides no specific engineering branches, constraints, or stage-gating. The LLM is free to add complexity in any way it deems appropriate, which may or may not be industrially relevant or logically coherent. From the resulting pool, we sampled 500 pairs.

- **Purpose:** This method directly tests our central hypothesis: that the *principled, guided, and staged* evolution of PLC-Spec-Syn is superior to generic, unconstrained complexity growth for creating high-quality training data.

**One-Shot Expert Prompt (OSEP-Corpus)**    This baseline represents the strongest possible non-evolutionary approach. It is designed to test whether a single, meticulously engineered "expert prompt" can generate complex, multi-faceted specifications that rival those produced by our multi-stage evolutionary process.

- **Process:** We crafted a highly-detailed prompt that instructed the LLM to act as a "Senior Control Systems Architect." For each of our 347 root specifications, the prompt required the LLM to generate a new, comprehensive specification that *simultaneously* integrated multiple features in a single step: a core functional extension, a critical safety interlock, a performance optimization (e.g., signal debouncing), and a maintenance feature (e.g., a cycle counter).

- **Purpose:** This method contrasts the "generate-all-at-once" paradigm with the "build-step-by-step" philosophy of PLC-Spec-Syn. It helps determine whether the value of our framework lies in the final complexity or in the logical coherence instilled by the gradual, cumulative evolutionary process itself.

**Modular Composition (MC-Corpus)**    This baseline simulates how human engineers build complex systems: by combining smaller, trusted, pre-existing components (bottom-up), rather than evolving a single monolithic program (top-down).

- **Process:** We first established two component libraries: an "Atomic Blocks" library with specifications from Generations 0-1, and a "Developed Modules" library with specifications from Generations 2-3. We then used a stratified sampling strategy: half the corpus was created by combining two "Atomic Blocks", and the other half by combining one "Atomic Block" with one "Developed Module". For each pair, an LLM acting as a "Systems Integrator" was prompted to generate a new, logically coherent specification that meaningfully combined the two components. From this pool of composite tasks, we sampled 500 pairs.

- **Purpose:** This method tests a "bottom-up" path to complexity, directly contrasting with PLC-Spec-Syn's "top-down" evolutionary approach. It helps determine if models learn better from tasks that are explicitly composed of simpler, known sub-problems, which may more realistically reflect large-scale industrial project development.

## T  VISUAL AND QUANTITATIVE COMPARISON AGAINST BASELINE METHODS

To complement the data presented in the main paper, this section provides a detailed visual and quantitative comparison of the corpora generated by our PLC-Spec-Syn framework against the five alternative baseline methodologies.

**Corpus Complexity and Richness.**    Figure 9 illustrates the average richness of the specifications. It clearly shows that the later stages of PLC-Spec-Syn (G5-6) produce the richest specifications, and our overall average surpasses all baselines except OSEP. Figure 10 demonstrates that this translates to more complex code, with PLC-Spec-Syn showing a controlled increase in both overall and logical (cyclomatic) complexity that surpasses simpler baselines.

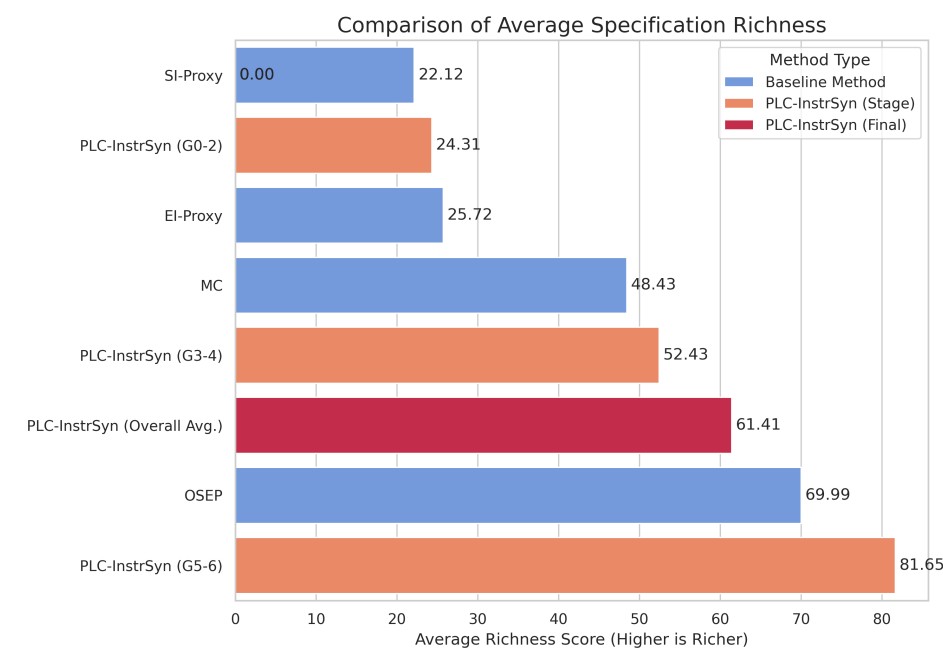

Figure 9: Comparison of the average specification richness score across methods.

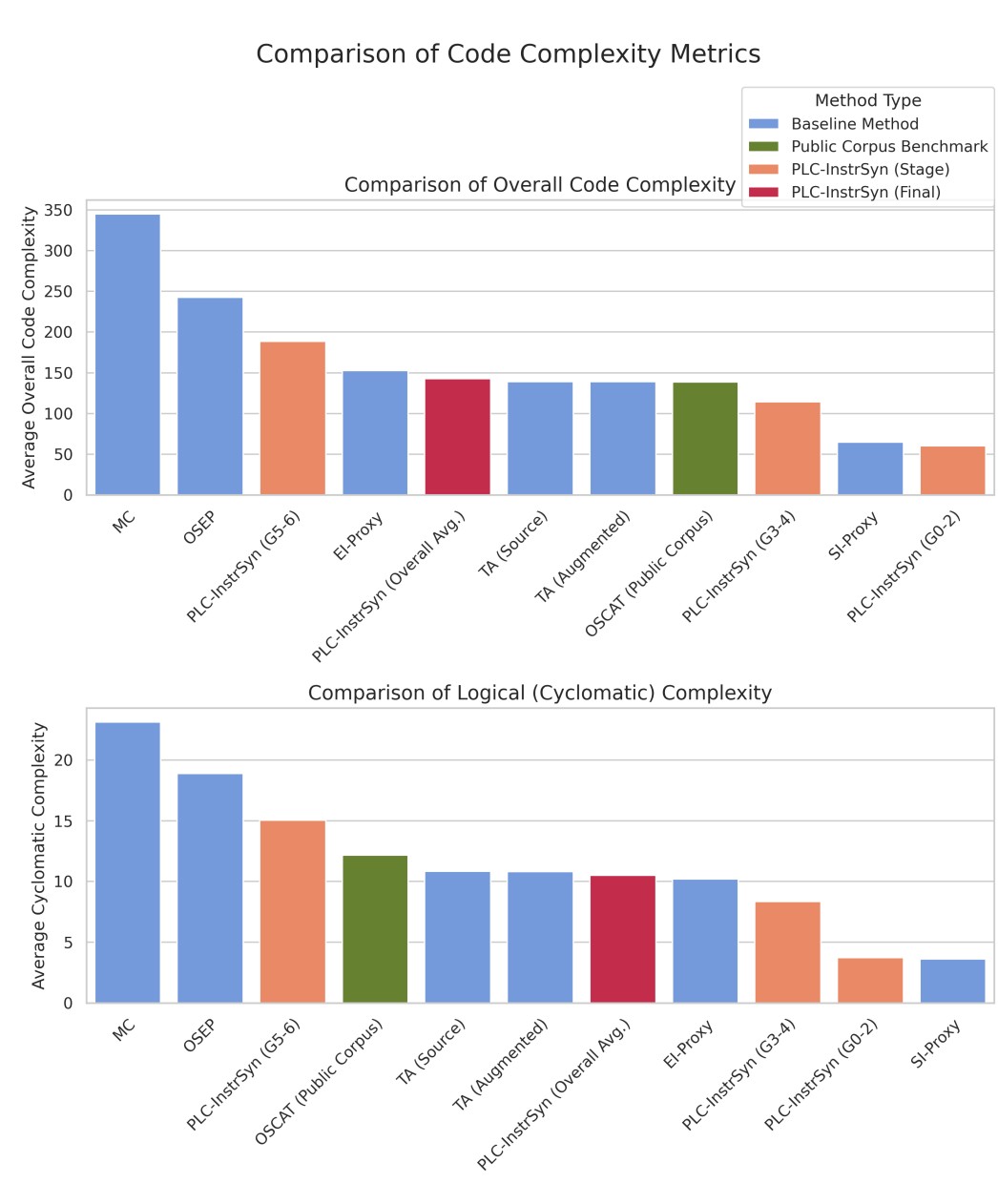

Figure 10: Comparison of code complexity metrics for all methods.

**Syntactic Diversity.** Beyond complexity, we analyzed the syntactic coverage of each generated corpus against the ST standard and the OSCAT library. As shown in Table 17, PLC-Spec-Syn (64.0%) achieves a superior balance of coverage, significantly outperforming most baselines and filling critical gaps in areas like `Control Flow Statements` where OSCAT is weakest. The Traditional Augmentation (TA-Corpus) method shows a slight increase in overall coverage from its source (64.0% to 66.3%) due to syntactic rewrites like loop swapping, but this change is superficial and does not represent the generation of semantically novel tasks.

**Richness vs. Generation Success Rate Trade-off.** To better understand the trade-offs inherent in different data generation paradigms, we analyzed task complexity against generation reliability. Figure 11 visualizes this critical trade-off, showing that PLC-Spec-Syn is unique in its ability to

Table 17: Syntactic coverage (%) of generated corpora, with OSCAT as a baseline. Our method demonstrates a superior balance of meaningful coverage. Best in category is **bold**.

| Feature Family | TA-Corpus (Source) | TA-Corpus (Augmented) | SI-Proxy | EI-Proxy | OSEP-Corpus | MC-Corpus | PLC-Spec-Syn (Ours Sample) | OSCAT (Baseline) |
|---|---|---|---|---|---|---|---|---|
| DataTypes (Elem. & Comp.) | 64.3% | 64.3% | 28.6% | 50.0% | 42.9% | 35.7% | 64.3% | **85.7%** |
| Control Flow Statements | 61.9% | **71.4%** | 23.8% | 42.9% | 42.9% | 38.1% | 61.9% | 9.5% |
| Standard Function Blocks | 55.6% | 55.6% | 66.7% | **77.8%** | **77.8%** | 55.6% | 55.6% | 22.2% |
| Operators (All Types) | 85.0% | 85.0% | 45.0% | 85.0% | 75.0% | 75.0% | 85.0% | 14.3% |
| **Overall Coverage** | 64.0% | **66.3%** | 36.0% | 55.1% | 51.7% | 51.7% | 64.0% | 29.2% |

produce tasks with high specification richness while maintaining a high success rate. It consistently operates in the desirable "High Richness, High Success" quadrant. In contrast, simpler baselines like SI-Proxy are confined to the "Low Richness" area, while more complex one-shot methods like OSEP and MC suffer from a low success rate.

The quantitative data supporting this visualization is provided in Table 18. The results clearly illustrate that our staged, guided evolutionary process is unique in its ability to generate progressively richer tasks without a drastic drop in generation success, a key challenge for unguided or one-shot generation methods.

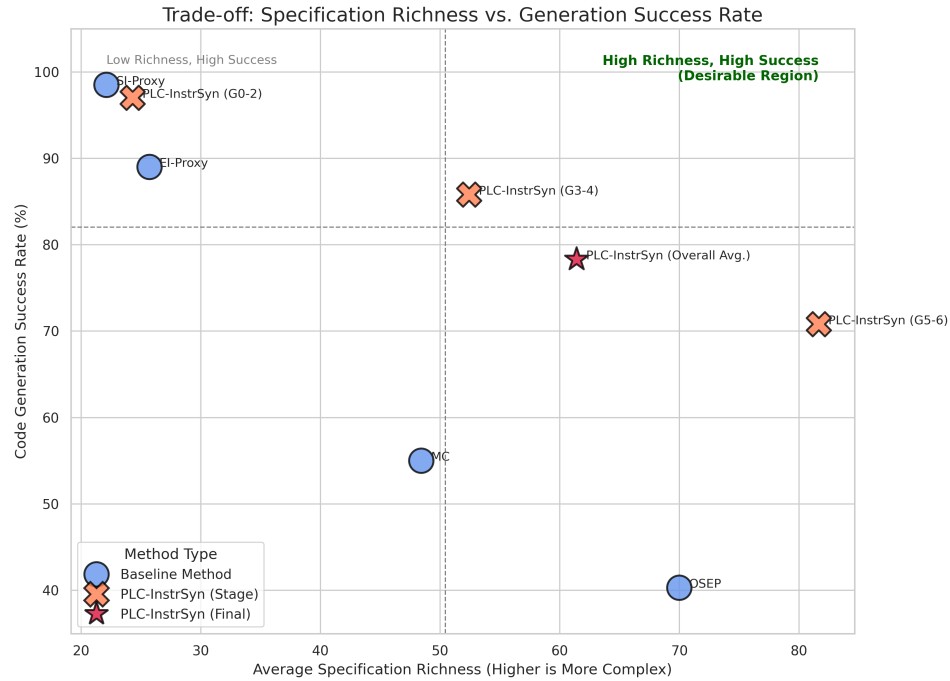

Figure 11: Trade-off between specification richness and generation success rate. PLC-Spec-Syn (coral 'X' marks and the red star) uniquely occupies the desirable top-right quadrant, demonstrating a superior balance of complexity and reliability.

Table 18: Comparison of Code Generation Success Rate against the average richness of the input specifications. PLC-Spec-Syn maintains a high success rate while generating progressively richer and more complex tasks.

| **Data Generation Method** | Avg. Spec Richness Score | Code Generation Success Rate (%) |
|---|---|---|
| *Baseline Methods* | | |
| Self-Instruct Proxy (SI) | 22.12 | 98.5% |
| Evol-Instruct Proxy (EI) | 25.72 | 89.0% |
| One-Shot Expert Prompt (OSEP) | 69.99 | 40.3% |
| Modular Composition (MC) | 48.43 | 55.0% |
| *PLC-Spec-Syn (Our Method) by Stage* | | |
| PLC-Spec-Syn (Gen 0-2, Early) | 23.32 | 97.0% |
| PLC-Spec-Syn (Gen 3-4, Mid) | 52.38 | 85.8% |
| PLC-Spec-Syn (Gen 5-6, Late) | 81.65 | 70.8% |
| **PLC-Spec-Syn (Overall Avg.)** | **58.55** | **78.3%** |

Table 19: Compilation success rate (%) on public benchmarks after fine-tuning on small, **500-sample** corpora from different generation methods. Results show the number of successful compilations and the success rate (%). Our method, PLC-Spec-Syn, consistently yields the highest performance. A detailed analysis of the baseline corpora is provided in Appendix T.

| Fine-tuning Corpus (500 samples) | Qwen2.5-Coder-1.5B-Instruct | | | | Qwen2.5-Coder-7B-Instruct | | | | Llama-3.2-1B-Instruct | | | |
| | OSCAT | LGF | Comp. | A4PLC | OSCAT | LGF | Comp. | A4PLC | OSCAT | LGF | Comp. | A4PLC |
|---|---|---|---|---|---|---|---|---|---|---|---|---|
| None (Base Model) | 91 (12.7%) | 7 (4.6%) | 6 (13.3%) | 81 (43.3%) | 108 (15.0%) | 20 (13.2%) | 8 (17.8%) | 96 (51.3%) | 16 (2.2%) | 3 (2.0%) | 0 (0.0%) | 5 (2.7%) |
| TA-Corpus | 95 (13.2%) | 7 (4.6%) | 6 (13.3%) | 82 (43.9%) | 111 (15.5%) | 20 (13.2%) | 8 (17.8%) | 97 (51.9%) | 18 (2.5%) | 3 (2.0%) | 0 (0.0%) | 5 (2.7%) |
| SI-Proxy | 110 (15.3%) | 10 (6.6%) | 7 (15.6%) | 88 (47.1%) | 125 (17.4%) | 23 (15.2%) | 9 (20.0%) | 104 (55.6%) | 30 (4.2%) | 6 (4.0%) | 1 (2.2%) | 11 (5.9%) |
| EI-Proxy | 121 (16.9%) | 12 (7.9%) | 7 (15.6%) | 92 (49.2%) | 136 (18.9%) | 25 (16.6%) | 9 (20.0%) | 109 (58.3%) | 41 (5.7%) | 8 (5.3%) | 1 (2.2%) | 15 (8.0%) |
| OSEP-Corpus | 128 (17.8%) | 14 (9.3%) | 8 (17.8%) | 95 (50.8%) | 144 (20.1%) | 27 (17.9%) | 10 (22.2%) | 113 (60.4%) | 49 (6.8%) | 10 (6.6%) | 2 (4.4%) | 18 (9.6%) |
| MC-Corpus | 135 (18.8%) | 16 (10.6%) | 8 (17.8%) | 99 (52.9%) | 153 (21.3%) | 29 (19.2%) | 10 (22.2%) | 116 (62.0%) | 58 (8.1%) | 12 (7.9%) | 2 (4.4%) | 22 (11.8%) |
| **PLC-Spec-Syn (Ours)** | **153 (21.3%)** | **22 (14.6%)** | **9 (20.0%)** | **105 (56.1%)** | **167 (23.3%)** | **31 (20.5%)** | **11 (24.4%)** | **120 (64.2%)** | **69 (9.6%)** | **15 (9.9%)** | **3 (6.7%)** | **28 (15.0%)** |

## U THE UNIFIED SPECIFICATION FORMAT

A core component of the PLC-Spec-Syn framework is the enforcement of a rigid, unified output format for every generated task specification. This structured template is not merely for consistency; it is engineered to be both highly machine-readable for our automated pipeline and representative of the detailed Functional Design Specifications (FDS) used in real-world industrial automation projects.

**Design Principles and Benefits.** The format was designed around several key principles that make it superior for generating high-quality, verifiable data:

- **Unambiguity by Separation:** The format strictly separates the natural language 'Control Logic Description' from the 'Formal Logic Expression'. This provides two views of the system's logic: a human-readable narrative for context and a mathematically precise set of equations that eliminates ambiguity, which is crucial for the formal verification stage.

- **Built-in Verifiability:** The inclusion of explicit 'Verification Requirements' in the form of test scenarios means that every specification is generated with its own ground-truth test plan. This directly enables and simplifies the downstream formal verification process, ensuring that the generated code can be checked for semantic equivalence against a clear set of criteria.

- **Traceability and Provenance:** Sections like 'System Metadata', 'Generation', 'Branch', and the 'Change Log' embed full traceability into each specification. This mimics the version control practices of industrial software engineering and allows for a complete audit of any task's evolutionary history.

- **Engineering Realism:** The structure mirrors authentic industrial documentation. Dedicated sections for 'I/O Requirements', 'Safety Requirements', and 'HMI/SCADA Mapping' reflect the standard components of a real control narrative, ensuring that the generated tasks are grounded in practical, real-world engineering considerations.

- **Forced Self-Correction:** The template concludes with mandatory 'Self-check' and 'Complexity Audit' sections. This forces the generating LLM to review its own output for compliance with our rules (e.g., monotonic complexity) before finalizing its response, acting as a powerful first-pass quality filter.

**Full Specification Template.** The complete template, including the strict rules enforced on the generating LLM, is provided below.

```
**UNIFIED OUTPUT FORMAT (use this exact structure)**

**CRITICAL DECLARATION HYGIENE RULES (MANDATORY):**
- Complete Variable Declaration: Every variable used must be declared.
- No Hidden Variables: All working variables must be listed in Internals.
- Cross-Reference Verification: Logic variables must match I/O list exactly.

# Industrial Control Specification

## System Metadata
```

```
- Asset: [Asset ID]
- Location: [Location]
- ... (Generation, Branch, Summary, etc.)

## Control Objective
[High-level goal of the control task]

## I/O Requirements
### Input Variables (CUMULATIVE)
- VarName: TYPE - Description
...
### Output Variables (CUMULATIVE)
- VarName: TYPE - Description
...
### Internals (CUMULATIVE)
- VarName: TYPE - Description
...
### Counts Check
`Inputs: X (+Y) | Outputs: X (+Y) | Internals: X (+Y)`

## Control Logic Description
[Natural language description of the control behavior]

**Formal Logic Expression**:
[Unambiguous boolean or timer-based equations]

## Safety Requirements
- [Critical safety interlock or condition]
...

## HMI/SCADA Mapping
| Tag Name | Type | R/W | Description |
|----------|------|-----|-------------|
| VarName  | TYPE | R/W | Description |

## Verification Requirements
### Test Scenarios
1. [Test Name]: [Inputs] -> [Expected Outputs]
...

## Self-check Summary & Complexity Audit
[LLM self-evaluation against rules like SEP-Compliance]
...
```

## V EXAMPLE OF A FULL EVOLUTION LINEAGE (SEP-COMPLIANT)

This appendix provides a detailed, generation-by-generation example of the Evo-Spec pipeline. The following evolution of a conveyor belt system has been carefully constructed to highlight the key principles of our methodology while adhering strictly to the Smooth Evolution Pacing (SEP) rules.

---

### V.1 GEN 0: SEED SPECIFICATION

```
# Industrial Control Specification
## System Metadata
- Asset: CON-042
- Branch: seed
## Control Objective
Turn ON PkgLine_Conveyor_Run when Opc_Start_PB is active.
## I/O Requirements
### Input Variables
- Opc_Start_PB: BOOL - Manual start pushbutton at operator console
### Output Variables
- PkgLine_Conveyor_Run: BOOL - Conveyor motor contactor signal
### Internals
(None)
### Counts
Inputs: 1 | Outputs: 1 | Internals: 0
## Control Logic Description
The conveyor motor is activated directly by the operator's start pushbutton.
Formal Logic Expression:
PkgLine_Conveyor_Run := Opc_Start_PB;
```

---

### V.2 GEN 1: FUNCTIONALITY

**Key Addition:** A master enable switch is added. (SEP: Early, +1 Input)

```
# Industrial Control Specification
## System Metadata
- Generation: 1 (early)
- Branch: functionality
## I/O Requirements (Cumulative)
### Input Variables
- Opc_Start_PB: BOOL - Manual start pushbutton
- PkgLine_Sys_Enable: BOOL - Master enable for the line (* NEW *)
### Output Variables
- PkgLine_Conveyor_Run: BOOL - Conveyor motor contactor signal
### Internals
(None)
### Counts Check
Inputs: 2 (+1) | Outputs: 1 (+0) | Internals: 0 (+0)
## Control Logic Description
The conveyor motor runs only when the system is enabled
AND the start button is pressed.
Formal Logic Expression:
PkgLine_Conveyor_Run := Opc_Start_PB AND PkgLine_Sys_Enable;
```

---

### V.3   GEN 2: SAFETY

**Key Addition:** A normally-closed Emergency Stop is introduced. (SEP: Early, +1 Input)

```
# Industrial Control Specification
## System Metadata
- Generation: 2 (early)
- Branch: safety
## I/O Requirements (Cumulative)
### Input Variables
- Opc_Start_PB: BOOL - Manual start pushbutton
- PkgLine_Sys_Enable: BOOL - Master enable for the line
- PkgLine_EStop_NC: BOOL - Emergency stop button (NC contact) (* NEW *)
### Output Variables
- PkgLine_Conveyor_Run: BOOL - Conveyor motor contactor signal
### Internals
(None)
### Counts Check
Inputs: 3 (+1) | Outputs: 1 (+0) | Internals: 0 (+0)
## Control Logic Description
The Emergency Stop button provides a high-priority override.
The conveyor can only run if the E-Stop is not active
(i.e., its normally-closed contact is TRUE).
Formal Logic Expression:
PkgLine_Conveyor_Run := Opc_Start_PB AND PkgLine_Sys_Enable
                        AND PkgLine_EStop_NC;
```

---

### V.4   GEN 3: PERFORMANCE

**Key Addition:** A 50ms timer is added to debounce the start button. (SEP: Mid, +1 Internal)

```
# Industrial Control Specification
## System Metadata
- Generation: 3 (mid)
- Branch: performance
## I/O Requirements (Cumulative)
### Input Variables
- Opc_Start_PB: BOOL - Manual start pushbutton
- PkgLine_Sys_Enable: BOOL - Master enable for the line
- PkgLine_EStop_NC: BOOL - Emergency stop button (NC contact)
### Output Variables
- PkgLine_Conveyor_Run: BOOL - Conveyor motor contactor signal
### Internals
- Opc_Start_Debounce_TON: TON - 50ms debounce timer (* NEW *)
### Counts Check
Inputs: 3 (+0) | Outputs: 1 (+0) | Internals: 1 (+1)
## Control Logic Description
The start pushbutton signal is filtered by a 50ms on-delay timer
to prevent false activations. The conveyor runs based on the
timer's output, while the E-Stop remains a direct, unfiltered override.
Formal Logic Expression:
VAR
    Opc_Start_Debounce_TON: TON;
END_VAR
Opc_Start_Debounce_TON(IN:=Opc_Start_PB, PT:=T#50ms);
PkgLine_Conveyor_Run := Opc_Start_Debounce_TON.Q
                        AND PkgLine_Sys_Enable
```

```
                                    AND PkgLine_EStop_NC;
```

---

## V.5 GEN 4: MAINTENANCE

**Key Addition:** A non-intrusive cycle counter with edge detection. (SEP: Mid, +2 Internals)

```
# Industrial Control Specification
## System Metadata
- Generation: 4 (mid)
- Branch: maintenance
## I/O Requirements (Cumulative)
### Input Variables
- Opc_Start_PB: BOOL - Manual start pushbutton
- PkgLine_Sys_Enable: BOOL - Master enable for the line
- PkgLine_EStop_NC: BOOL - Emergency stop button (NC contact)
### Output Variables
- PkgLine_Conveyor_Run: BOOL - Conveyor motor contactor signal
### Internals
- Opc_Start_Debounce_TON: TON - 50ms debounce timer
- PkgLine_Mtr_CycleCount: UINT - Motor cycle counter (* NEW *)
- PkgLine_LastRunState: BOOL - Previous run state for edge detection (* NEW *)
### Counts Check
Inputs: 3 (+0) | Outputs: 1 (+0) | Internals: 3 (+2)
## Control Logic Description
The primary control logic is unchanged. A parallel, non-intrusive
maintenance logic increments PkgLine_Mtr_CycleCount each time
the conveyor transitions from stopped to running.
Formal Logic Expression:
VAR
    Opc_Start_Debounce_TON: TON;
    PkgLine_Mtr_CycleCount: UINT;
    PkgLine_LastRunState: BOOL;
END_VAR
(* Main Logic *)
Opc_Start_Debounce_TON(IN:=Opc_Start_PB, PT:=T#50ms);
PkgLine_Conveyor_Run := Opc_Start_Debounce_TON.Q
                        AND PkgLine_Sys_Enable
                        AND PkgLine_EStop_NC;
(* Maintenance Logic - Rising edge detection *)
IF (PkgLine_Conveyor_Run AND NOT PkgLine_LastRunState) THEN
    PkgLine_Mtr_CycleCount := PkgLine_Mtr_CycleCount + 1;
END_IF;
PkgLine_LastRunState := PkgLine_Conveyor_Run;
```

---

## V.6 GEN 5: INTEROPERABILITY

**Key Addition:** A remote status bit for SCADA integration. (SEP: Late, +1 Output)

```
# Industrial Control Specification
## System Metadata
- Generation: 5 (late)
- Branch: interoperability
## I/O Requirements (Cumulative)
### Input Variables
- Opc_Start_PB: BOOL - Manual start pushbutton
```

```
2322   - PkgLine_Sys_Enable: BOOL – Master enable for the line
2323   - PkgLine_EStop_NC: BOOL – Emergency stop button (NC contact)
2324   ### Output Variables
2325   - PkgLine_Conveyor_Run: BOOL – Conveyor motor contactor signal
2326   - SCADA_Conveyor_Status: BOOL – Remote status bit for SCADA (* NEW *)
2327   ### Internals
2328   - Opc_Start_Debounce_TON: TON – 50ms debounce timer
2329   - PkgLine_Mtr_CycleCount: UINT – Motor cycle counter
2330   - PkgLine_LastRunState: BOOL – Previous run state for edge detection
2331   ### Counts Check
       Inputs: 3 (+0) | Outputs: 2 (+1) | Internals: 3 (+0)
2332   ## Control Logic Description
2333   The control logic is unchanged. A status output is added for SCADA
2334   monitoring, mirroring the conveyor's run status for remote visibility.
2335   Formal Logic Expression:
2336   VAR
2337       Opc_Start_Debounce_TON: TON;
2338       PkgLine_Mtr_CycleCount: UINT;
2339       PkgLine_LastRunState: BOOL;
2340   END_VAR
2341   (* Main Logic – unchanged *)
2342   Opc_Start_Debounce_TON(IN:=Opc_Start_PB, PT:=T#50ms);
2343   PkgLine_Conveyor_Run := Opc_Start_Debounce_TON.Q
2344                          AND PkgLine_Sys_Enable
                              AND PkgLine_EStop_NC;
2345   (* Maintenance Logic – unchanged *)
2346   IF (PkgLine_Conveyor_Run AND NOT PkgLine_LastRunState) THEN
2347       PkgLine_Mtr_CycleCount := PkgLine_Mtr_CycleCount + 1;
2348   END_IF;
2349   PkgLine_LastRunState := PkgLine_Conveyor_Run;
2350   (* Interface Logic – NEW *)
2351   SCADA_Conveyor_Status := PkgLine_Conveyor_Run;
2352
2353
```

---

```
2354
2355   V.7   GEN 6: CONTEXTUAL COMPLICATION
2356
2357   Key Addition: Adaptive cleaning mode with jog operation. (SEP: Late, +1 Input, +1 Internal)
2358   # Industrial Control Specification
2359   ## System Metadata
2360   - Generation: 6 (late)
2361   - Branch: contextual_complication
2362   ## I/O Requirements (Cumulative)
2363   ### Input Variables
2364   - Opc_Start_PB: BOOL – Manual start pushbutton
2365   - PkgLine_Sys_Enable: BOOL – Master enable for the line
2366   - PkgLine_EStop_NC: BOOL – Emergency stop button (NC contact)
2367   - Maint_CleanMode_Sel: BOOL – Cleaning mode selector switch (* NEW *)
       ### Output Variables
2368   - PkgLine_Conveyor_Run: BOOL – Conveyor motor contactor signal
2369   - SCADA_Conveyor_Status: BOOL – Remote status bit for SCADA
2370   ### Internals
2371   - Opc_Start_Debounce_TON: TON – 50ms debounce timer
2372   - PkgLine_Mtr_CycleCount: UINT – Motor cycle counter
2373   - PkgLine_LastRunState: BOOL – Previous run state for edge detection
2374   - Clean_Jog_TON: TON – 10ms timer for cleaning mode jog (* NEW *)
2375   ### Counts Check
       Inputs: 4 (+1) | Outputs: 2 (+0) | Internals: 4 (+1)
```

```
## Control Logic Description
A cleaning mode provides safe jog control for maintenance.
When selected, the conveyor uses a shorter debounce (10ms)
for responsive manual control during cleaning operations.
Formal Logic Expression:
VAR
    Opc_Start_Debounce_TON: TON;
    Clean_Jog_TON: TON;
    PkgLine_Mtr_CycleCount: UINT;
    PkgLine_LastRunState: BOOL;
END_VAR
(* Main Logic with Cleaning Mode *)
IF Maint_CleanMode_Sel THEN
    (* Cleaning mode: Short debounce for jog control *)
    Clean_Jog_TON(IN:=Opc_Start_PB, PT:=T#10ms);
    PkgLine_Conveyor_Run := Clean_Jog_TON.Q
                        AND PkgLine_Sys_Enable
                        AND PkgLine_EStop_NC;
ELSE
    (* Normal mode: Standard debounced operation *)
    Opc_Start_Debounce_TON(IN:=Opc_Start_PB, PT:=T#50ms);
    PkgLine_Conveyor_Run := Opc_Start_Debounce_TON.Q
                        AND PkgLine_Sys_Enable
                        AND PkgLine_EStop_NC;
END_IF;
(* Maintenance Logic - counts both modes *)
IF (PkgLine_Conveyor_Run AND NOT PkgLine_LastRunState) THEN
    PkgLine_Mtr_CycleCount := PkgLine_Mtr_CycleCount + 1;
END_IF;
PkgLine_LastRunState := PkgLine_Conveyor_Run;
(* Interface Logic *)
SCADA_Conveyor_Status := PkgLine_Conveyor_Run;
```

# W    USE OF LLMS

The LLM's role was strictly limited to improving the quality of the prose. Its contributions included:

- Assisting with grammar, spelling, and punctuation.
- Rephrasing sentences and paragraphs for better clarity, conciseness, and flow.
- Helping to structure and format content, including generating LaTeX code for tables.

