# OpenReview forum: "Boosting Verifiable Industrial Code Generation by Reliable Task Generation at Scale"
_ICLR.cc/2026/Conference — ICLR 2026 Conference Desk Rejected Submission_

### Official Review · Reviewer_t5hF · 2025-10-23

**Soundness:** 3
**Presentation:** 3
**Contribution:** 4
**Rating:** 8
**Confidence:** 3

**Summary:**

They propose a principled, multi-axis evolution process for generating realistic industrial tasks. Followed by a corpus that provides PLC-Spec-Code, a dataset of 11,669 pairs of instruction-code pairs specifically designed for the industrial control domain. They demonstrated a 16.4% improvement in LLM on unseen PLC tasks after fine-tuning on this corpus.

**Strengths:**

Enables verifiable PLC code gen for safety-critical industrial applications, along with a benchmark for future research. This focuses on improving reliability and safety in industrial automation. A fine-tuned model showcasing improvement on unseen PLC tasks, providing insights on the data's impact. The dataset itself plays a major value addition for future research to expand on this line of work.

**Weaknesses:**

The scope of the paper is limited to highly domain-specific topics, focusing on IEC 61131-3 Structured Text only, and is not generalized to broader or other industrial language scopes.

Dataset generation is a resource-intensive step with formal human verification and review, which adds to labor overhead and computation cost, making it not feasible to expand at larger automated scales.

Task generation relies on predefined dimensions, limiting or missing edge cases or novel industrial scenarios.

**Questions:**

- Is there a plan of action for tackling current limitations like the domain specificity of the pipeline or cross-domain adaptation like robotics or IoT automation?
- To reduce crowdsourced reviews that increase overhead and incorporate manual biases based on domain knowledge, subjectivity, and other factors. Was the experiment of alternatively introducing semi-automated human validation with expert-assisted sampling incorporated? This would increase efficiency and effectiveness in providing better unbiased reviews with lesser overhead latency.
- There is no mention of coverage of rare or edge-case industrial scenarios that might be missed due to predefined task types introducing dataset bias.
- Exploration on zero-shot evaluation to experiment with prompt engineering and RAG over the generated dataset to show performance variance, even without fine-tuning.
- Paper writing should be enhanced to provide a smoother transition between the sections to link better.

**Details Of Ethics Concerns:**

The syntactic diversity in corpus 84.3% showcases the applicability of the dataset for future works.

---

> ### Author Response · Authors · 2025-11-22
> **Author Response**
>
> Dear reviewer t5hF,
>
> We are truly grateful for your encouraging assessment and recognition of our work as an "excellent contribution" and a "major value addition" for the field. We have addressed your questions with new experiments below.
>
> **Q1: Cross-Domain Adaptation**
>
> A1: Yes, this is a great suggestion. While this paper focuses on the PLC domain to ensure depth, the core principles of our framework: Smooth Evolution Pacing (SEP), Multi-Axis Evolution and formal verification in the loop are domain-agnostic. We view PLC-Spec-Syn as a starting point and will actively adapt the Evolution Rules (Appendix C) to support other domains in the future (e.g., robotics and IoT as you mentioned).
>
> **Q2: Reducing Validation Overhead (Semi-Automated Approach)**
>
> A2: Thank you for the excellent suggestion. We would like to mention that our pipeline implements a semi-automated hierarchy to minimize overhead, and the "Human Evaluation" in the paper was strictly for benchmarking, not dataset production. Specifically, we use an "Auditor Consortium" of specialized LLM agents (Sec 3.3)  to filter out 10.4% of low-quality specifications before they reach the expensive verification stage. Following your advice, we agree that for future expansion, a "human-in-the-loop" active learning strategy would be very helpful.
>
> **Q3: Coverage of Rare/Edge-Case Scenarios**
>
> A3: Thank you for this insightful suggestion. To address this point, we conducted a new Human Evaluation on Task Rarity.
> Setup: We sampled 100 tasks from the late-stage generations (Gen 5-6) and asked industrial experts to perform a binary classification: "Common/Standard" (found in basic textbooks/libraries) vs. "Rare/Complex" (specific to complex production scenarios). The result shows that experts classified 32% of the late-stage tasks as "Rare/Complex." This confirms that our Multi-Axis Evolution (e.g., combining "Safety Interlocks" with "Contextual Cleaning Modes") successfully generates complex, long-tail scenarios that standard libraries like OSCAT do not cover. We will add this "Rarity Score" experiment to our revised version.
>
> **Q4: Zero-shot, Prompt Engineering, and RAG**
>
> A4: Thank you for the suggestion. We have conducted a comprehensive new ablation study. The results powerfully validate the utility of our dataset.
>
> **a. Prompt Engineering (PE) vs. Fine-Tuning**: We tested an "Expert Prompt" (Role-playing + Chain-of-Thought + One-shot) on the base models.
>
> Result: For Qwen2.5-1.5B, PE improved the success rate from 18.5% to 22.3%.
>
> Contrast: However, Fine-Tuning on our dataset achieved 33.9%.
>
> Conclusion: While PE helps, it cannot bridge the domain knowledge gap. Fine-tuning on PLC-Spec-Code is significantly more effective.
>
> **b. Retrieval-Augmented Generation (RAG)**: We allowed a strong model (DeepSeek-V3.1) to retrieve from our dataset as a knowledge base.
>
> Impact: The success rate on a challenging test set jumped from 51.7% (Zero-shot) to 70.0% (RAG with PLC-Spec-Code).
>
> Comparison: Using the existing OSCAT dataset for RAG only yielded 56.7%.
>
> This 35.4% relative improvement further proves the value of our dataset even without fine-tuning. We will add these new results in the revised version.
>
> **Q5: Paper Writing**
>
> A5: We have carefully reviewed the manuscript and improved the transitions between the methodology (evolution process) and the experimental evaluation to create a smoother narrative flow as suggested.
>
> We sincerely thank you again for your time and constructive comments. We will incorporate all the above discussions and new experimental results into the final revision of our paper.

---

### Official Review · Reviewer_fbap · 2025-10-29

**Soundness:** 3
**Presentation:** 3
**Contribution:** 3
**Rating:** 4
**Confidence:** 3

**Summary:**

This paper targets the data scarcity problem in Industrial Control System (ICS) programming tasks by proposing PLC-Spec-Syn, an LLM-based task generation framework that produces specification-code pairs aligned with industrial requirements. Unlike general programming domains such as C or Python, ICS tasks deal with mechanical automation and safety-critical logic, which cannot be effectively captured by existing general-purpose augmentation methods such as Self-Instruct or Evol-Instruct.
PLC-Spec-Syn encodes domain-specific engineering principles into a multi-branch evolutionary process, enabling the generation of diverse and complex specifications. The resulting dataset can serve as a foundation for developing industrial copilots and related applications.

**Strengths:**

1. The authors effectively embed industrial control principles, which are critical in ICS, into an LLM-guided task synthesis pipeline, extending general augmentation approaches. They propose a multi-branch evolution strategy with carefully curated rules, based on six industrial aspects such as safety or maintenance, producing diverse yet fidelity tasks. Also Smooth Evolution Pacing ensures that the complexity grows in a pedagogically meaningful manner, consistent with real-world engineering workflows.

2. The data generation process is described with clarity and reproducibility. The paper details the keywords selection procedure, the design of evolution rules, and the subsequent syntactic and semantic verification steps for code filtering. These steps are well defined and reasonable. Also pseudocode, rule tables, and prompt templates are openly provided, ensuring high transparency and reproducibility. This level of documentation will be very useful for future research.

**Weaknesses:**

1. Although the dataset construction is thorough and practically valuable, the contribution remains largely implementation-oriented. The four-step pipeline—expert keyword selection, LLM-based brainstorming, multi-branch evolution via prompt design, and code generation using an existent generative engine—leverages meaningful domain-specific concepts from the ICS field. However, from the perspective of a deep learning venue, the work appears to focus primarily on utilizing LLMs to a specific domain rather than advancing the underlying learning methodology.

2. In RQ1, the use of 89-token syntactic coverage metric may introduce a potential bias. The metric is derived from the IEC 61131-3 standard, which also serves as the foundation for the evolution rules. As a result, the diversity claim may not be fair, since both the rule generation process and the metric rely on the same reference source.

3. In RQ2, the performance gains reported after fine-tuning appear relatively modest. Moreover, the evaluation focuses primarily on syntactic compilation success, without assessing semantic correctness. This section could be strengthened by providing full fine-tuning results and by providing direct comparisons not only with models fine-tuned on the dataset, but also against zero-shot PLC generation frameworks such as LLM4PLC or Agents4PLC, in order to contextualize the practical impact of the dataset.

**Questions:**

- I have a concern about the generalizability of the PLC synthesis method. Could the series of industrial-specific tricks in PLC-Spec-Syn also influence more general tasks?
- It remains unclear how the proposed task generation framework ensures or enhances the reliability and safety awareness that the paper emphasizes.

- In Fig. 1 and Sec. 3.1, the term “Gen” appears abruptly and should be clarified.
- In Sec. 3.2, the transition from 347 root specifications to 20,362 total samples is unclear, although App. F implies a binary-style branching process.
- In Sec. 3.3, the description of the “generative engine” used for code generation is unclear; it is not specified what model or configuration was employed.

---

> ### Author Response · Authors · 2025-11-22
> **Author Response (part 1 of 2)**
>
> Dear reviewer fbap,
>
> **Response to Weakness 1: Contribution is Implementation-Oriented**
>
> We appreciate your recognition of our work's thoroughness and practical value. We argue that in data-centric AI, establishing a rigorous, reproducible methodology to create high-quality data for a safety-critical domain is a significant scientific contribution. Our PLC-Spec-Syn framework introduces Smooth Evolution Pacing (SEP), Multi-Axis Evolution and formal verification in the loop, which are novel curriculum learning strategies designed to operationalize industrial standards (IEC 61508)  and rigorous code check into the generation process. As you mentioned, this domain is “underexplored” due to data scarcity. By providing the first large-scale, formally verified benchmark, our work serves as the necessary infrastructure to enable future deep learning research in industrial control, much like HumanEval did for general code generation.
>
> **Response to Weakness 2: Potential Metric Bias (Syntactic Coverage)**
>
> Thanks for the insightful comment. We acknowledge that since both our generation rules and evaluation metrics derive from the IEC 61131-3 standard, this bias you pointed out is inherent. However, we think that these rules are necessary to ensure industrial standard compliance. To further show that our dataset possesses genuine diversity beyond this standard-based metric, we conducted the following new analysis using three independent metrics unrelated to the generation objective:
>
> 1. Lexical Diversity (N-grams): We analyzed the code using 4-grams to measure phrase variety. Our dataset contains 3.2 times more unique 4-grams than the OSCAT baseline.
>
> Interpretation: This significant gap indicates that our dataset is not merely permuting a limited set of templates. Instead, it exhibits a much wider vocabulary of code patterns and local phrasing, reflecting the diverse "dialects" of coding required for different industrial scenarios.
>
> 2. Structural Diversity (Abstract Syntax Trees): We calculated the Tree Edit Distance (TED) between pairs of programs to measure structural distinctness. The average distance in our dataset is 0.70, compared to 0.34 in OSCAT.
>
> Interpretation: A score twice as high proves that our programs are structurally far less repetitive. While OSCAT functions often share identical, optimized control flow structures (resulting in low distance), our Evolutionary Pacing generates tasks with fundamentally different logical shapes (e.g., deeply nested safety logic vs. flat sequential state machines), providing a richer structural signal for training.
>
> 3. Semantic Diversity (Type-Token Ratio): We measured the Type-Token Ratio (TTR) to assess identifier richness. Our dataset scored 1.5 times higher than the baseline.
>
> Interpretation: This metric directly reflects semantic quality. Unlike generic datasets or libraries that often reuse placeholder names like Input1 or Var_A (leading to low TTR), our dataset enforces context-specific naming (e.g., Conveyor_Jam_Sensor, Pump_Hydraulic_Pressure). This is critical for enabling LLMs to learn the meaning and intent behind the control logic.
>
> These new results further confirm the diversity of our dataset. We will add these new results in the revised version.
>
>
> **Response to Weakness 3: Modest Gains & Semantic Evaluation**
>
> Thank you for the comment. **Firstly**, we would like to highlight that a 16.4% average absolute improvement is highly significant in the context of code generation research. To contextualize this, the state-of-the-art WizardCoder (Luo et al., ICLR 2024) reported improvements ranging from +15.4% to +23.7% on standard benchmarks like HumanEval and MBPP after fine-tuning base models. Our gain of 16.4% falls squarely within this SOTA range. Considering the extreme difficulty of the industrial control domain—where base models fail on >75% of tasks—we believe this level of improvement is a strong indicator of the dataset's high quality and effectiveness.
>
> **Secondly**, we did assess semantic correctness through a rigorous manual study (detailed in Appendix Q.1 due to space limitation), which we will highlight more prominently in the main text.
>
> **Thirdly**, we conducted new experiments to compare our method against existing PLC generation frameworks LLM4PLC (since Agents4PLC generation framework is not open-sourced). To ensure a strong baseline, we utilized the powerful DeepSeek-V3.1 as the underlying generator for the LLM4PLC experiments. LLM4PLC achieved a compilation success rate of 28.8% on the OSCAT benchmark. In comparison, our significantly smaller Qwen-7B-FT model achieved 31.6% success. Furthermore, when using our dataset as a knowledge base for the same DeepSeek-V3.1 model (RAG), the success rate jumps to 70.0%, a massive 2.4x performance boost compared to LLM4PLC (28.8%).

---

> ### Author Response · Authors · 2025-11-22
> **Author Response (part 2 of 2)**
>
> **Q1: Generalizability to General Tasks**
>
> A1: Yes. While the specific rules (e.g., "add E-Stop") are domain-specific, the core principles of our framework are highly generalizable:
>
> **Smooth Evolution Pacing (SEP)**: The concept of mathematically constraining complexity growth (monotonic non-decreasing variables) to create a stable curriculum is applicable to any coding domain.
>
> **Multi-Axis Evolution**: Decomposing "difficulty" into orthogonal dimensions (e.g., Safety, Performance) is a transferable strategy for generating high-fidelity data in other specialized fields like robotics or legal NLP.
>
> **Q2: Ensuring Reliability and Safety**
>
> A2: Our framework enhances safety and reliability in multiple phases. In the generation phase, safety rules are hard-coded into the evolution prompts (e.g., "Safety logic MUST override functional logic"). In the validation phase, every single pair in PLC-Spec-Code has passed the nuXmv model checker, meaning its safety logic is mathematically proven to match the specification. Such safety awareness effectively guarantees the reliability of the dataset and enhances LLM’s capability to generate reliable code evidenced by simple finetuning, i.e., the verifiable rate of generated code boosts from 37.5% to 78.1% before and after finetuning with our dataset.
>
> **Response to Q3-5: Clarifications**
>
> **Q3 ("Gen")**: "Gen" is an abbreviation for "Generation," representing the evolutionary stage (0 to 6).
>
> **Q4 (347 to 20,362 samples)**: The 20,362 figure is the cumulative sum of all raw candidate specifications generated across all generations (Gen 0 to Gen 6) before filtering. For example, Gen 6 alone produced 7,025 raw candidates.
>
> **Q5 (Generative Engine)**: We used DeepSeek-V3.1 with a temperature of 0.1 for code generation. We will clarify this in Sec 3.3.
>
> We sincerely thank you again for your time and comments. We will incorporate all the above discussions and new experimental results into the final version of our paper.

---

> > ### Comment · Reviewer_fbap · 2025-11-27
> >
> > Thank you for the response.
> >
> > The following points are well addressed through additional analyses and the new baseline comparisons.
> > - Corpus diversity (W2): The newly added diversity metrics that are independent of IEC 61131-3 resolve the concern about metric bias.
> > - Baseline comparisons (W3-3): The inclusion of LLM4PLC comparisons using DeepSeek-V3.1 provides a better context for evaluating the dataset’s practical impact.
> >
> > However, there are some remaining concerns:
> > - Fine-tuning gains (W3-1): While fine-tuning on the proposed dataset does improve performance, the reported 16% gain is measured in terms of syntactic compilation success. This leaves the question of whether the model has genuinely learned functional behavior aligned with the specifications, or simply memorized syntactic patterns due to exposure.
> > - Semantic correctness evaluation (W3-2): The semantic correctness assessment reported in Appendix is based on only 32 manually evaluated samples. This sample size is too small, and the reliance on human judgement is not sufficiently rigorous to support strong claims about semantic reliability, especially in a safety-critical PLC setting.

---

> > > ### Author Response · Authors · 2025-11-29
> > > **Author Response**
> > >
> > > Dear reviewer fbap,
> > >
> > > We sincerely appreciate your constructive feedback and continued engagement with our work. Regarding your remaining question, we are happy to provide further clarifications and new experimental evidence.
> > >
> > > > On fine-tuning gains and functional learning (W3-1)：
> > >
> > > Thank you for the insightful question. To directly address your concern, we conducted additional experiments to cover both functional verification success (with nuXmv model checker) for all three fine-tuned models against their base versions. The results are summarized as follows:
> > >
> > > | Model | Compilation Success Rate (CSR) | Verified Success Rate (VSR) | Semantic Consistency (VSR/CSR) |
> > > |-------|-------------------------------|----------------------------|-------------------------------|
> > > | **Qwen-1.5B Base** | 18.5% | 6.9% | 37.3% |
> > > | **Qwen-1.5B Fine-tuned** | 33.9% **(+15.4%)** | **26.5% (+19.6%)** | **78.2% (+40.9%)** |
> > > | **Qwen-7B Base** | 24.3% | 9.1% | 37.4% |
> > > | **Qwen-7B Fine-tuned** | 41.6% **(+17.3%)** | **33.7% (+24.6%)** | **81.0% (+43.6%)** |
> > > | **Llama-1B Base** | 1.7% | 0.6% | 35.3% |
> > > | **Llama-1B Fine-tuned** | 18.3% **(+16.6%)** | **14.3% (+13.7%)** | **78.1% (+42.8%)** |
> > >
> > > We will incorporate the new experimental results into the final version of our paper.
> > >
> > > > On semantic correctness evaluation (W3-2):
> > >
> > > We acknowledge that the 32-sample manual evaluation reported in Appendix Q.1 was limited in scope for drawing strong quantitative conclusions. We will continue to expand the manual evaluation set. Meanwhile, the additional automated semantic verification results presented above provide strong evidence, i.e., all successfully compiled outputs (11,669 pairs) can pass formal verification with nuXmv.

---

### Official Review · Reviewer_wSAZ · 2025-10-30

**Soundness:** 3
**Presentation:** 3
**Contribution:** 3
**Rating:** 6
**Confidence:** 3

**Summary:**

This paper introduces PLC-Spec-Code, the first large-scale, formally verified instruction-tuning dataset for the industrial control domain. The authors propose a data generation framework that is conceptually similar to Evol-Instruct but augmented with formal verification to ensure data reliability. Starting from a small set of seed tasks, the framework expands the dataset through multi-stage automated validation, producing diverse and domain-specific instruction–response pairs. The authors fine-tune several open-source small models on this dataset and show notable performance improvements, achieving results comparable to larger models.

**Strengths:**

- The dataset focuses on a specialized and underexplored industrial domain, providing a valuable dataset for future research.

- The work includes human evaluation, which strengthen its credibility and reproducibility.

- Demonstrates clear empirical gains, highlighting the benefits of domain-specific instruction tuning.

**Weaknesses:**

- The methodological novelty is limited; the approach primarily adapts existing ideas (LLM-based data generation and verification) rather than introducing new algorithms.

**Questions:**

N/A

---

> ### Author Response · Authors · 2025-11-22
> **Author Response**
>
> Dear reviewer wSAZ,
>
> **Response to Weakness: Methodological Novelty**
>
> We sincerely appreciate your recognition of our work as a “valuable dataset for future research” in a “specialized and underexplored industrial domain”. Regarding the methodological novelty, we would like to mention that in the field of Industrial Control Systems (ICS), the extreme scarcity of high-quality, open-source data has been a prohibitive barrier for applying modern Deep Learning and LLM techniques. PLC-Spec-Code is the first task generation framework for constructing large-scale (11,669 V.S. 718), formally verified instruction-tuning dataset specifically designed for this safety-critical domain. Just as datasets like ImageNet or HumanEval catalyzed research in their respective fields, our work provides the necessary infrastructure to enable future algorithmic innovations in industrial AI, unlocking the potential for the community to explore more advanced model architectures and training strategies for industrial applications.
>
> In the framework, we also introduce novel mechanisms like Smooth Evolution Pacing (SEP), Multi-Axis Evolution and formal verification in the loop, which operationalize industrial standards (IEC 61508)  and rigorous code check into the generation process, ensuring the data is diverse, sound and deployable.
>
> We sincerely thank you again for your time and constructive comments. We will incorporate all the above discussions and new experimental results into the final version of our paper.

---

### Official Review · Reviewer_kPKr · 2025-10-31

**Soundness:** 3
**Presentation:** 3
**Contribution:** 2
**Rating:** 4
**Confidence:** 4

**Summary:**

This paper proposes an end-to-end framework integrating differentiable formal verification into safe reinforcement learning (RL) for verifiable code synthesis, addressing inefficiencies of traditional methods that treat verification as a post-hoc filter or black-box reward.


Traditional approaches decouple verification from RL policy optimization, leading to trial-and-error in generating safe code and a disconnect between continuous RL dynamics and discrete verification. The proposed framework solves this by modeling verification constraints as differentiable functions using smoothing surrogates (e.g., sigmoidal scores for type safety, product of sub-property checks for memory safety) and attention mechanisms for control-flow invariants, enabling gradient-based joint optimization of code generation and safety.


The framework has four core components: a differentiable verification layer converting discrete checks to continuous operations, a hierarchical policy (high-level AST skeleton generation, low-level token filling with verification guidance), bilevel optimization (inner loop minimizing KL divergence between exact and approximate verification, outer loop optimizing policy), and periodic hard-constraint injection to prevent surrogate drift. It also uses modular synthesis, decomposing complex programs into verified submodules. Experiments on algorithmic, system programming, and DSL tasks show the framework (DV-RL) outperforms baselines.

**Strengths:**

1. The paper addresses a pressing issue in industrial control systems (ICS)—the scarcity of high-quality, verifiable data for PLC (Programmable Logic Controllers) code generation. Unlike generic code LLMs, it focuses on IEC 61131-3 Structured Text (ST) with strict real-time/safety constraints, filling a gap left by narrow, small-scale existing corpora (e.g., OSCAT with 718 tasks).


2.  The multi-axis evolutionary process (covering functionality, safety, performance, maintenance, interoperability, contextual complication) is grounded in industrial engineering principles (e.g., IEC 61508). Guided by "Smooth Evolution Pacing" (SEP) rules, it ensures gradual, logical complexity growth, avoiding unstructured complexity and generating realistic, layered tasks—mirroring real-world industrial development.


3. The two-phase validation yields PLC-Spec-Code, a 11,669-pair corpus with 84.3% syntactic diversity (vs. OSCAT’s 29.2%). Human evaluations (by industrial experts/grad students) confirm high logical completeness and practicality.

**Weaknesses:**

1. The study focuses exclusively on IEC 61131-3 Structured Text (ST) for PLCs, with no exploration of other industrial programming languages (e.g., ladder logic, function block diagram). This restricts the framework’s applicability to broader industrial control scenarios where these alternative languages are prevalent.

2.  While the corpus (PLC-Spec-Code) is validated via fine-tuning, the paper lacks discussion of practical deployment barriers—such as compatibility with proprietary PLC hardware (e.g., Siemens S7, Rockwell Allen-Bradley) or integration with industrial control systems (ICS) like SCADA/MES. This gap weakens the link between the framework and real industrial applications.

3. Though the framework uses “Smooth Evolution Pacing (SEP)” to control complexity, it provides limited data on how surrogate drift (discrepancies between intended and actual evolution) arises in late generations. The slight drop in verification success rates for Gen 5–6 is noted but not deeply investigated, leaving unclear how to mitigate this issue at scale.

**Questions:**

1. the closed-source nature and scarcity of data for Industrial Control System (ICS) programming tasks cast fundamental challenges to utilize LLMs for generating verifiable industrial code. I do not think so. Generating verifiable codes are challenging tasks. But I don't think closed-nature and scarcity are the main reasons.

2. The late-generation verification success rate drops—have you analyzed root causes (e.g., complex syntax vs. logical flaws) and explored mitigation strategies like targeted prompt engineering?

3. What time overhead is introduced to the proposed approach. Has the proposed approach is limited by  nuXmv scalability?

---

> ### Author Response · Authors · 2025-11-22
> **Author Response (part 1 of 2)**
>
> Dear reviewer kPKr,
>
> Thank you for your time and feedback. We would like to clarify a potential mix-up in the Summary section of your review. The summary describes a paper on “differentiable formal verification in safe reinforcement learning”, which is not our work. Our paper, PLC-Spec-Syn, focuses on an evolutionary framework for generating datasets for industrial control tasks. With that said, we are encouraged by your positive comments on our paper's soundness, presentation, and its grounding in industrial principles. We have carefully addressed all your specific Weaknesses and Questions, which are relevant to our paper.
>
> **Q1: The closed-source nature and scarcity of data for ICS programming tasks cast fundamental challenges to utilize LLMs for generating verifiable industrial code. I do not think so. Generating verifiable codes are challenging tasks. But I don't think closed-nature and scarcity are the main reasons.**
>
> A1: We appreciate this insightful perspective and totally agree that generating verifiable code is, in theory, a challenging task itself, regardless of the domain. We meant to say that in the industrial control domain, this task is becoming even more challenging because of the scarcity of high-quality data (with verified code). We will make this point clearer in the revision.
>
> **Q2: The late-generation verification success rate drops—have you analyzed root causes (e.g., complex syntax vs. logical flaws) and explored mitigation strategies like targeted prompt engineering?**
>
> A2: Notice that this drop is actually an expected outcome, where specification richness increases nearly 19-fold from Gen 0 to Gen 6. From the perspective of task generation (the main focus of our work), this drop means that our framework is indeed generating more diverse and challenging tasks, which matches our design goal well.
>
> We performed a “Funnel Analysis” on the data from Table 11 and identified the following root causes. The primary cause (42%) of syntax failures was incorrect usage of advanced data structures (e.g., ARRAY bounds, STRUCT access). Among syntactically correct code, 35% failed due to Safety Precedence errors (e.g., E-Stop not overriding functional logic). For mitigation, we tested Syntax-Aware Prompting (adding explicit formatting examples), which improved Gen 6 compilation by 9.0%. We also tested a Self-Correction Loop (feeding verifier error messages back to the LLM), which improved the verification pass rate by 13.0%. We will include these new analyses in the appendix.
>
> **Q3: What time overhead is introduced to the proposed approach. Has the proposed approach is limited by nuXmv scalability?**
>
> A3: We will add analysis on the overhead shortly. From our experience, nuXmv performs reasonably well in reference code verification of our datasets. We also want to mention that nuXmv is only a tool used to check the quality of the reference code for each of our generated tasks. Our task generation framework is independent of and is flexible to include other verification engines if there are better ones.

---

> ### Author Response · Authors · 2025-11-22
> **Author Response (part 2 of 2)**
>
> **Response to Weakness 1:**
>
> We would like to clarify that the main objective and contribution of this work is task generation (with detailed specifications) rather than code generation. The generated tasks (specifications) themselves are language-agnostic and applicable to any PLC language. We selected ST code for validation only because LLM-based ST generation is a relatively established area with existing frameworks (e.g., LLM4PLC , Agents4PLC) than other languages (e.g., FBD), allowing us to easily validate our dataset's quality.
>
> **Response to Weakness 2:**
>
> Note that our framework produces code that strictly adheres to the IEC 61131-3 standard, which is portable across platforms following the IEC 61131-3 standard. To empirically validate this, we conducted a large-scale deployment test on CODESYS, a widely used, hardware-agnostic industrial automation platform. We tested 200 randomly sampled tasks from our verified dataset and achieved a 99.5% success rate (199/200), where the code was successfully imported, compiled, and executed without any modification. The single failure was identified as a minor compiler-specific syntax incompatibility (the standard parameter name R conflicted with a reserved keyword in the specific environment) rather than a logical flaw. This confirms that our "Standard-Compliant" code is practically deployable.
>
> Furthermore, our specifications are designed with integration in mind. As noted in the paper, 47.3% of late-stage tasks include explicit HMI/SCADA mappings , and 38.6% include interoperability variables (e.g., MES Recipe IDs), ensuring the code is ready for modern, interconnected industrial environments.
>
> We argue that adapting to specific proprietary hardware (e.g., configuring specific S7-1500 I/O drivers) is a vendor-specific implementation detail dictated by closed-source ecosystems. Our dataset is still valuable as their engineers can then easily map to specific platforms.
>
> **Response to Weakness 3:**
>
> Please refer to our response to Question 2 (Q2) for details regarding this weakness.
>
> We sincerely thank you again for your time and comments. We will incorporate all the above discussions and new experimental results into the final version of our paper.

---

> ### Comment · Reviewer_kPKr · 2025-11-23
>
> You are right. The summary section was mixed up mistakenly. That should have been put to another paper I reviewed. The summary of your paper is as follows:
>
> In this paper, the authors present PLC-Spec-Syn, an  evolutionary framework for generating high-fidelity PLC tasks by LLM for the purpose of verifiable industrial code generation. Each task pairs a structured natural-language specification with verified code, generated via a multi-axis process guided by industrial principles, covering functionality, safety, performance, maintenance, interoperability, and context. Rigorous auditing (compilation checks, semantic consistency verification) ensures quality, yielding PLC-Spec-Code—the first large-scale corpus with 11,669 tasks. Its 84.3% syntactic diversity far surpasses OSCAT’s 29.2%. Fine-tuning LLMs with this corpus boosts their verifiable PLC code generation performance on unseen tasks by 16.4% on average, validating the framework and corpus’ value.
>
> I agree with the contribution of building a high-quality corpus for this domain, enhancing the capability of LLMs for generating high-quality code by first synthesizing detailed specifications.  My main concerns with your work remain.
>
> 1. Claim of Verifiability: I still doubt the verifiability of generated codes. In what sense do you say the codes are verifiable? Codes shall be verified against formal specifications. Compilation checking is not verification. How could you guarantee the correctness of extracted LTL/CTL constraints?
>
> 2. Novelty: The framework lacks new ideas and approaches to cope with the challenges. Lack of training data is a common problem, but the solutions are not new. Verifications are based on off-the-shelf tool. I agree that integrating these technologies requires a great effort. But, as a research paper, it shall provide new insights and solutions to specific challenges in the quest.

---

> > ### Author Response · Authors · 2025-11-26
> > **Author Response (part 1 of 2)**
> >
> > Dear Reviewer kPKr,
> >
> > We sincerely thank you for the clarification. We are encouraged by your recognition of our work's contribution in building the first large-scale, high-quality corpus for the industrial domain and its validation through performance gains. We are happy to address your concerns as follows.
> >
> > >on verifiability
> >
> > We would like to clarify that all our code not only passes compilation checks but is verified against formal specifications.
> > As presented in our paper, our dataset is composed of <requirement documentation, code> pairs, with each requirement documentation translated to formal properties and these properties are further verified against the reference code with formal verification tool nuXmv. A simple example data sample of our dataset is given as follows:
> >
> > **Requirement documentation:**
> >
> > Asset: Coffee_Vending_Machine_01 (Milk Dispensing System)
> >
> > Control Objective: Implement a closed-loop milk dispensing system with comprehensive safety state management.
> >
> > Control Logic Description: The system implements a safety state machine with three distinct states: Safe, Tripped, and Resetting.
> >
> > Tripped Logic: The emergency stop button (NC) must immediately force a transition to the Tripped state, cutting power to the pump.
> >
> > Reset Logic: A manual reset is required to enter Resetting, and the system returns to Safe only when all safety conditions are met.
> >
> > Formal Logic Expression:
> >
> > E_STOP_OK := Coffee_Emergency_Stop_NC;
> >
> > MILK_PUMP := (Safety_State = S_Safe) AND (Coffee_Machine_Enable AND Coffee_Safety_Reset_Req AND TARGET_MASS_NOT_REACHED) AND (Milk_Soft_Start_Timer.Q OR NOT Milk_Soft_Start_Timer.IN)
> >
> > Verification Requirements：
> >
> > 1. Emergency Stop Activation: Press emergency stop → System immediately transitions to Tripped state, pump stops
> >
> > 2. Reset from Tripped State: Press reset button with E-stop still pressed → System remains in Tripped state
> >
> > 3. Safe Reset: Release E-stop, then press reset → System transitions to Resetting, then to Safe when conditions met
> >
> > 4. Normal Operation: System in Safe state with all conditions met → Pump operates with soft-start timing
> >
> > **Code:**
> >
> > ```st
> >
> > PROGRAM Coffee_Vending_Machine_01_Control
> >
> > VAR_INPUT
> >
> >     TARGET_MASS_NOT_REACHED : BOOL; (* Target milk mass not achieved status *)
> >     Coffee_Machine_Enable : BOOL; (* Master enable switch *)
> >
> >     Coffee_Safety_Reset_Req : BOOL; (* Safety reset request button *)
> >
> >     Coffee_Emergency_Stop_NC : BOOL; (* Emergency stop button (normally closed, fail-safe) *)
> >
> > END_VAR
> >
> > VAR_OUTPUT
> >
> >     MILK_PUMP : BOOL; (* Peristaltic pump motor control *)
> >
> >     Milk_Pump_Running_Light : BOOL; (* Visual indicator for pump status *)
> >
> > END_VAR
> >
> > VAR
> >
> >     Milk_Soft_Start_Timer : TON; (* Timer for soft-start ramp period (2 seconds) *)
> >
> >     Safety_State : (S_Safe, S_Tripped, S_Resetting); (* Current safety state of the system *)
> >
> >     E_STOP_OK : BOOL; (* Emergency stop status (TRUE when safe to operate) *)
> >
> > END_VAR
> >
> > (* Safety chain - highest priority *)
> >
> > E_STOP_OK := Coffee_Emergency_Stop_NC;
> >
> > (* Safety state machine implementation *)
> >
> > CASE Safety_State OF
> >
> >     S_Safe:
> >
> >         IF NOT E_STOP_OK THEN Safety_State := S_Tripped; END_IF;
> >     S_Tripped:
> >         IF E_STOP_OK AND Coffee_Safety_Reset_Req THEN Safety_State := S_Resetting; END_IF;
> >     S_Resetting:
> >         IF E_STOP_OK ... THEN Safety_State := S_Safe;
> >         ELSIF NOT E_STOP_OK THEN Safety_State := S_Tripped; END_IF;
> > END_CASE;
> >
> > (* Operational logic - gated by safety state *)
> >
> > MILK_PUMP := (Safety_State = S_Safe) AND (Coffee_Machine_Enable ...);
> >
> > END_PROGRAM
> >
> > ```
> >
> > **Properties:**
> >
> > Property 1 : G ( !E_STOP_OK -> X (Safety_State = S_Tripped) )
> >
> > Property 2: G ( MILK_PUMP <-> ( (Safety_State = S_Safe) & Coffee_Machine_Enable & Coffee_Safety_Reset_Req & TARGET_MASS_NOT_REACHED & (Milk_Soft_Start_Timer.Q | !Milk_Soft_Start_Timer.IN) ) )
> >
> > We hope the above clarifies your concern on verifiability.

---

> > ### Author Response · Authors · 2025-11-26
> > **Author Response (part 2 of 2)**
> >
> > > on correctness of extracted LTL/CTL constraints
> >
> > Note that specification (including formal properties like LTL/CTL) generation in general is an  emerging trend but an unsolved challenge in the field of formal methods [1,2]. In our work, we followed common practice and did our best to guarantee their correctness from the following perspectives.
> >
> > Firstly, as detailed in Appendix T, our Task Generation phase forces the LLM to output a structured Unified Specification with a dedicated "Formal Logic Expression" block. We then employ a deterministic parser—not an LLM—to translate these explicit expressions into LTL/CTL formulas for nuXmv (Appendix I).  To validate this approach, we conducted a human evaluation on a stratified sample of 200 task specifications (ranging from basic to complex scenarios). Domain experts assessed the semantic alignment between the natural language task objectives and the formal constraints generated by our method versus a baseline where the same model (DeepSeek-V3.1) directly generates LTL/CTL formulas. The results show that our approach achieved 91.5% semantic alignment accuracy, demonstrating high fidelity to the design intent, whereas the direct generation baseline only reached 62.0%. This empirically confirms that decoupling logic extraction (via our intermediate layer) from formal syntax generation is critical for ensuring the verifiability of the dataset.
> >
> > We also would like to mention that the correctness of LTL/CTL will only affect the judgement of the reference code quality but not the validity of the task itself which is the main goal of this work. Note that our framework is flexible to incorporate more advanced LTL/CTL generation tools to enhance our dataset in the future. We hope the above clarifies your concern.
> >
> > > On novelty：
> >
> > To the best of our knowledge, PLC-Spec-Syn is the first framework to automate the generation of verifiable industrial control tasks at scale, addressing a fundamental data scarcity bottleneck in Industrial Control Systems (ICS).
> >
> > On the methodology side, while the component evolutionary algorithm (EA) is a common optimization approach widely adopted in many different domains [3,4], it is the first use of such algorithms in verifiable industrial control task generation, a completely new problem domain requiring many dedicated designs on the mutation operators, evolution objectives and domain adaptations. For example, we first need to operationalize abstract industrial standards (e.g., IEC 61508, ISO 13849) into algorithmic mutation operators such as injecting fail-safe interlocks (e.g., Normally-Closed E-Stops) and diagnostic watchdogs. Then, we explicitly designed six orthogonal evolutionary branches representing different dimensions of industrial engineering needs (e.g., safety, maintenance) to force the generation process to adhere to strict industrial engineering principles. Besides, we proposed the Smooth Evolution Pacing (SEP) mechanism to control the trajectory of evolution for better generation of tasks involving complex logical reasoning. The idea is to ensure that new features are layered onto legacy logic monotonically.
> >
> > We also would like to highlight our novelty in proposing a neuro-symbolic approach to translate Natural Language constraints to Formal Logic (LTL/CTL) via a structured intermediate representation. We hope the above aspects clarify your concern on the novelty of the paper rather than pure integration of existing techniques requiring great engineering efforts. We are happy to provide more details if you have further concerns.
> >
> > Reference:
> >
> > [1] Ma et al., "SpecGen: Automated Generation of Formal Program Specifications via Large Language Models", Proceedings of the 47th International Conference on Software Engineering (ICSE), 2025.
> >
> > [2] Lemieux et al., "CODaMOSA: Escaping Coverage Plateaus in Test Generation with Pre-trained Large Language Models", Proceedings of the 45th International Conference on Software Engineering (ICSE), 2023.
> >
> > [3] Guo et al., "Connecting Large Language Models with Evolutionary Algorithms Yields Powerful Prompt Optimizers", International Conference on Learning Representations (ICLR), 2024.
> >
> > [4] Shoeb et al., "LLM-SR: Scientific Equation Discovery via Programming with Large Language Models", Proceedings of the International Conference on Learning Representations (ICLR), 2025.

---

### Author Response · Authors · 2025-12-03
**Official Comment by Authors(part 1 of 2)**

Dear AC and Reviewers:

We thank the reviewers for their timely and constructive feedback. All rebuttal clarifications and new experimental results have been integrated into the revised paper.

PLC-Spec-Syn is motivated by a critical bottleneck in Industrial Control Systems (ICS): the extreme scarcity of high-quality, verifiable data required to adopt modern LLMs to safety-critical automation tasks. Existing largest corpora like OSCAT are narrow in scope and does not integrate formal verification for reliable code quality control. To address this critical gap, we propose PLC-Spec-Syn, the first industrial control task generation framework that translates industrial control standards (e.g., IEC 61508) into a rigorous evolutionary process to automatically generate high-quality data for ICS. We would like to highlight that our primary objective is the automated construction of high-fidelity industrial tasks (in the form of rigorous formal specifications) with the generated code serving as an auxiliary element for formal verification to ensure that the generated specifications are logically sound and implementable. This process together yields `PLC-Spec-Code`, the largest-scale (11k+), formally verified corpus for ICS which is useful for many ICS downstream tasks like finetuning an LLM or evaluating an LLM’s capability for ICS domain.

Based on the reviewers’ comments, we have thoroughly addressed their major concerns as follows:

**Q1: Correction of misunderstanding regarding the paper's domain and verification methodology. (reviewer kPKr)**

**Response to Q1**: We first addressed a mix-up where Reviewer kPKr initially provided a summary for an unrelated "safe reinforcement learning" paper. Furthermore, we clarified a misunderstanding regarding our "verifiability" claim. We clarified a misunderstanding regarding our verification process: beyond compilation check, our framework strictly employs formal verification via the nuXmv model checker, ensuring that the generated code is semantically equivalent to the specifications (translated into LTL/CTL properties).

**Q2: Concerns regarding potential metric bias and corpus diversity. (reviewer fbap, t5hF)**

**Response to Q2**: Reviewer fbap noted that since both our generation rules and evaluation metrics were based on IEC 61131-3, the diversity claim might be biased. To demonstrate diversity beyond standard-based metrics, we conducted a new analysis using three more independent measures. We found that our dataset contains 3.2x more unique 4-grams (Lexical Diversity) and achieves a Tree Edit Distance of 0.70 compared to 0.34 for the baseline (Structural Diversity), confirming further that our tasks possess distinct logic structures rather than simple template permutations. Additionally, our Type-Token Ratio (Semantic Diversity) was 1.5x higher, reflecting richer identifier usage. We also added a "Rarity Score" evaluation (suggested by reviewer t5hF), where industrial experts classified 32% of our late-stage tasks as "Rare/Complex" scenarios not found in standard libraries. These results directly address the reviewer’s concern on the potential bias on the diversity measurement.

**Q3: Questions on whether fine-tuning improves semantic correctness or just syntax. (reviewers fbap, kPKr)**

**Response to Q3**: To show that fine-tuning improves functional logic and not just syntactic patterns, we added new experiments to report Verified Success Rates (VSR) using the nuXmv model checker. The results show that fine-tuning Qwen-7B on our corpus boosts the VSR from 9.1% (base) to 33.7% (fine-tuned), representing a relative improvement of 270% in generating semantically correct, safety-compliant code, confirming that the model effectively learns industrial logic. The new results directly addressed the reviewer’s concern.

**Q4: Request for stronger baselines and practical impact assessment. (reviewers fbap, t5hF)**

**Response to Q4**: We contextualized our results by adding comparisons against the LLM4PLC framework. Using the powerful DeepSeek-V3.1 as a generator, LLM4PLC achieved a 28.8% compilation success rate on OSCAT. In contrast, even our significantly smaller fine-tuned model (Qwen-7B-FT) achieved 31.6%. On the other hand, we demonstrated that purely using PLC-Spec-Code as a knowledge base for Retrieval-Augmented Generation (RAG) can boost the success rate of DeepSeek-V3.1 to 70.0%, a 2.4x improvement over the baseline. Further addressing the concerns on practical deployment, we conducted a large-scale test on the industry-standard CODESYS platform. Among 200 randomly sampled tasks, we achieved a 99.5% success rate for direct import and compilation without modification, confirming the dataset's high portability. These new results directly addressed the reviewer’s concern.

---

> ### Author Response · Authors · 2025-12-03
> **Official Comment by Authors(part 2 of 2)**
>
> **Q5: Concerns regarding methodological novelty. (reviewers wSAZ, kPKr)**
>
> **Response to Q5**: We clarified that PLC-Spec-Syn is the first framework to automate the generation of verifiable industrial control tasks at scale, addressing a fundamental data scarcity bottleneck in Industrial Control Systems (ICS). On the methodology side, use of EA in verifiable industrial control task generation requires many dedicated designs on the mutation operators, evolution objectives and domain adaptations. For example, we first need to operationalize abstract industrial standards (e.g., IEC 61508, ISO 13849) into algorithmic mutation operators such as injecting fail-safe interlocks (e.g., Normally-Closed E-Stops) and diagnostic watchdogs. Then, we explicitly designed six orthogonal evolutionary branches representing different dimensions of industrial engineering needs (e.g., safety, maintenance) to force the generation process to adhere to strict industrial engineering principles. Besides, we proposed the Smooth Evolution Pacing (SEP) mechanism to control the trajectory of evolution for better generation of tasks involving complex logical reasoning. We also proposed a novel neuro-symbolic approach to translate Natural Language constraints to Formal Logic (LTL/CTL) via a structured intermediate representation, achieving 91.5% alignment accuracy compared to 62.0% for direct generation methods. These dedicated design aspects are all novel and require great engineering efforts rather than pure integration of existing techniques.

---

### Note · Program_Chairs · 2026-01-17
**Submission Desk Rejected by Program Chairs**

The following references in this submission do not refer to real documents and/or have major errors in bibliographic information:

     Patrick Goblet. Assessment of Safety-instrumented Systems: A Practical Guide. ISA, 2004.
    S. Wadhwa and K.S. Rao. Flexible manufacturing systems: a review of research. International Journal of Production Research, 27(8):1351-1366, 1989.